# Identification of osteoclast-osteoblast coupling factors in humans reveals links between bone and energy metabolism

Megan M. Weivoda[1,2], Chee Kian Chew[1,3], David G. Monroe[1], Joshua N. Farr [1], Elizabeth J. Atkinson[1], Jennifer R. Geske [1], Brittany Eckhardt[1], Brianne Thicke[1], Ming Ruan[1], Amanda J. Tweed[1], Louise K. McCready[1], Robert A. Rizza[1], Aleksey Matveyenko[1], Moustapha Kassem[4,5], Thomas Levin Andersen[4,5], Adrian Vella[1], Matthew T. Drake[1], Bart L. Clarke[1], Merry Jo Oursler[1]* & Sundeep Khosla [1]*

Bone remodeling consists of resorption by osteoclasts followed by formation by osteoblasts, and osteoclasts are a source of bone formation-stimulating factors. Here we utilize osteoclast ablation by denosumab (DMAb) and RNA-sequencing of bone biopsies from post-menopausal women to identify osteoclast-secreted factors suppressed by DMAb. Based on these analyses, *LIF, CREG2, CST3, CCBE1*, and *DPP4* are likely osteoclast-derived coupling factors in humans. Given the role of Dipeptidyl Peptidase-4 (DPP4) in glucose homeostasis, we further demonstrate that DMAb-treated participants have a significant reduction in circulating DPP4 and increase in Glucagon-like peptide (GLP)-1 levels as compared to the placebo-treated group, and also that type 2 diabetic patients treated with DMAb show significant reductions in HbA1c as compared to patients treated either with bisphosphonates or calcium and vitamin D. Thus, our results identify several coupling factors in humans and uncover osteoclast-derived DPP4 as a potential link between bone remodeling and energy metabolism.

[1] Mayo Clinic College of Medicine and Science, Rochester, MN, USA. [2] University of Michigan School of Dentistry, Ann Arbor, MI, USA. [3] Tan Tock Seng Hospital, Singapore, Singapore. [4] University of Southern Denmark, Odense, Denmark. [5] Odense University Hospital, Odense, Denmark. *email: Oursler. merryjo@mayo.edu; khosla.sundeep@mayo.edu

During bone remodeling, bone resorption by osteoclasts is followed by bone formation by osteoblasts[1]. These processes are balanced through coupling to ensure the precise formation of new bone at sites of resorption to maintain skeletal mass and strength throughout the lifespan; however, with aging and certain diseases, resorption exceeds bone formation, leading to bone loss and increased risk for osteoporotic fractures[2,3].

Because of the excess bone resorption contributing to bone loss in these conditions, anti-resorptive therapies, for example, bisphosphonates and denosumab (DMAb), have become the first-line treatments for osteoporosis[4]. While these agents effectively reduce resorption and limit further bone loss, the resultant decrease in osteoclasts is associated with a coupling-related decrease in osteoblast number and bone formation rate, thus limiting the efficacy of anti-resorptive therapies[5]. In addition, bone anabolic therapies, such as teriparatide, have a significantly blunted effect to stimulate bone formation when given following treatment with anti-resorptive drugs[6,7], suggesting that the presence of osteoclasts is necessary for certain bone anabolic therapies[8].

The effect of osteoclasts to promote osteoblast-mediated bone formation is also evidenced in the human genetic disease, osteopetrosis. Osteopetrosis is a condition of high bone mass caused by mutations in genes involved in osteoclast differentiation or function, and patients exhibit increased bone mass that is brittle due to impaired removal of old bone[9,10]. There are several forms of osteopetrosis, which are classified by mutations, inheritance pattern, and effects on either osteoclast differentiation or function. Osteopetrosis caused by mutations in genes necessary for osteoclast differentiation is characterized by decreased osteoclast numbers. Similar to patients on traditional anti-resorptive therapy, osteopetrosis patients with decreased osteoclast numbers also exhibit decreased osteoblast numbers and bone formation rates. In contrast, osteopetrosis patients with mutations affecting osteoclast resorptive activity, rather than differentiation, such as mutations in *CLCN7* or *TCIRG1*, exhibit normal or increased osteoclast numbers and maintained/increased bone formation rates, leading to more severe, debilitating forms of osteopetrosis[9,10].

In contrast to traditional anti-resorptive therapies, odanacatib, a small-molecule inhibitor of cathepsin K (CTSK), blocks bone resorption without reducing osteoclast numbers. Similar to osteopetrosis patients with impaired osteoclast resorptive activity, clinical trials of odanacatib showed promise to reduce resorption with minimal decreases in bone formation rate[11–15]. However, further development of this therapy was impeded by an off-target increase in cerebrovascular accident incidence in odanacatib-treated patients[16]. While this particular therapy did not progress, the data provide proof of concept that pharmacological approaches designed to inhibit osteoclast activity, rather than osteoclast numbers, is a viable approach to prevent bone loss without reducing bone formation. In addition, further elucidation of mechanisms by which osteoclasts promote bone formation may yield additional therapeutic targets to stimulate bone formation even in the absence of osteoclasts.

In the present study, we utilized a single DMAb treatment as a biological probe to pharmacologically ablate osteoclasts in postmenopausal women in order to identify potential osteoclast-derived factors that contribute to the coupling of bone resorption and bone formation in humans. Our findings independently confirm leukemia inhibitory factor (LIF) as a coupling factor in human bone remodeling. In addition, we identify dipeptidyl peptidase-4 (DPP4) as an osteoclast-derived factor that may link bone remodeling and receptor activator of nuclear factor kappa b ligand (RANKL) signaling to energy metabolism, and provide a mechanism for a possible improvement in glycemic control in diabetic individuals following treatment with anti-RANKL therapies.

## Results

**Effect of DMAb on serum markers of resorption and formation.** A total of 56 postmenopausal women were assessed for eligibility (Supplementary Fig. 1). Fifty-two of these women were randomly assigned to a one-time treatment of placebo or DMAb. Of the 26 participants assigned to placebo, 25 received the intervention. All 26 participants assigned to DMAb received the intervention. One participant in the placebo arm, and two participants in the DMAb arm, withdrew following intervention. Thus, final biopsy, serum/plasma collection, and analysis were performed on 24 participants per group.

Participants were well matched for baseline clinical characteristics (Fig. 1a). As demonstrated in Fig. 1b, participant baseline C-terminal telopeptide of type I collagen (CTX), a serum marker of bone resorption, correlated significantly with the serum markers of bone formation, procollagen type 1 N-terminal propeptide (P1NP, $R = 0.72$, $p < 0.0001$; Spearman's rank correlation) and osteocalcin (OCN, $R = 0.72$, $p < 0.0001$; Spearman's rank correlation). Participant baseline serum tartrate-resistant acid phosphatase 5b (TRAP5b), a marker of osteoclast number, also significantly correlated with P1NP and OCN levels ($R = 0.43$, $p = 0.002$; $R = 0.31$, $p = 0.031$, respectively; Spearman's rank correlation), although to a lesser degree than CTX. Three months post treatment, peripheral blood serum, bone marrow plasma, and needle bone biopsies were collected, as described previously[17,18]. As expected, participants treated with DMAb exhibited a significant reduction in serum markers of bone resorption (CTX and TRAP5b), as well as a coupling-related decrease in bone formation markers (P1NP and OCN; Fig. 1c).

**Identification of osteoclast and osteoblast gene signatures.** The bone was centrifuged to remove loosely adherent marrow; the remaining bone sample (bone and cells on the bone surface, e.g., osteoclasts, osteoblasts, lining cells, etc.) was homogenized for RNA isolation. RNA samples with the highest yield and quality were then submitted for RNA-sequencing (RNA-seq) ($N = 15$ per group; thus, analyses utilizing sequencing data are limited to $N = 15$, and sample sizes for other analyses are specified as appropriate). Sequencing data was then used to determine genes that were significantly altered in DMAb-treated participants. Nine hundred forty-eight genes were significantly changed by DMAb in the centrifuged bone ($p < 0.05$; R package edgeR), with 128 exhibiting a false discovery rate below 0.10.

A comprehensive list of bone-related genes was compiled from the Gene Ontology (GO) Biological Processes Database, with processes including Bone Development, Bone Mineralization, Bone Morphogenesis, Bone Remodeling, Bone Resorption, Bone Trabecula Formation, Ossification, Osteoblast Differentiation, and Osteoclast Differentiation Biological Processes (Supplementary Table 1). These processes represented a total of 555 bone-related genes with overlap between multiple gene sets; of these genes, 481 were represented in the centrifuged bone biopsy RNA-seq results, with 48 genes significantly altered in centrifuged bone samples from the DMAb-treated participants. Of these, 40 genes were significantly decreased in bone samples from the DMAb-treated participants, whereas 8 genes were significantly upregulated (Supplementary Table 2).

Eight of the DMAb-downregulated genes were selected as accepted markers of osteoclasts (*CTSK*, *CALCR*, *SIGLEC15*, *ACP5*, *DCSTAMP*, *OCSTAMP*, *TNFRSF11A*, and *MMP9*), and eight as accepted markers of the osteoblast lineage (*COL1A1*, *SPARC*, *SP7*, *SPP1*, *BGLAP*, *IBSP*, *ALPL*, and *GJA1*). In assessing a heat map of these significantly decreased osteoclast and osteoblast genes in placebo vs. DMAb-treated participants, it was clear that placebo participants exhibiting high or low

**Fig. 1 Baseline participant characteristics and effects of DMAb on bone metabolism. a** Baseline clinical and biochemical data in the study participants. **b** Correlations at baseline between markers of bone formation (PINP, μg/L; OCN, ng/mL) and resorption (CTX, ng/mL; TRAP5b, U/L) demonstrating coupling of bone formation and resorption at the systemic level ($N = 48$ participants); Spearman's correlation coefficient was used to determine strength of relationships. **c** Changes in markers of bone resorption (top panels) and bone formation (bottom panels) over 3 months (% change from baseline, $N = 24$ participants per group). Individual values are plotted with mean and error bars represent SD. Significance was determined using the Mann–Whitney test. Source data are provided as a Source Data file.

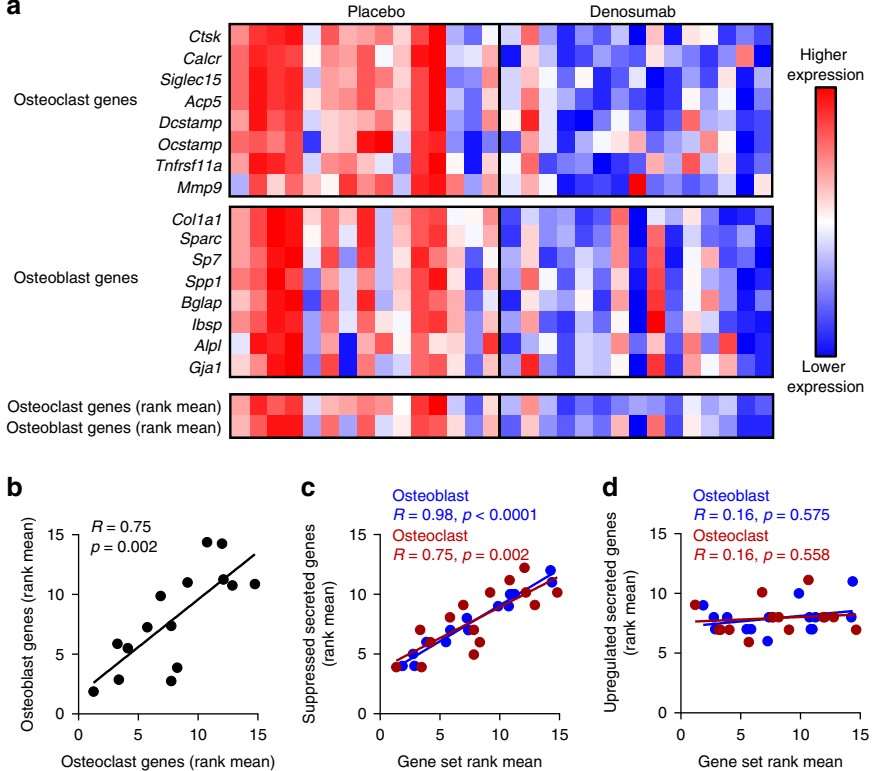

**Fig. 2 Correlation of osteoclast and osteoblast genes and secreted factors altered by DMAb. a** Heat maps showing the osteoclast and osteoblast normalized gene expression in placebo and DMAb-treated participant bone biopsies. Normalized gene expression (CQN values) were ranked for each gene across the placebo and DMAb participant biopsies ($N = 15$ participant biopsies/group). Red denotes higher expression and blue denotes lower expression. **b** Rank mean values for DMAb-suppressed osteoclast and osteoblast gene sets were plotted for the placebo participants. **c** DMAb-suppressed secreted factor genes, but not **d** DMAb-upregulated secreted factor genes correlate with osteoblast and osteoclast marker genes in the placebo participants ($N = 15$ participant biopsies); Spearman's correlation coefficient was used to test the strength of the relationship between rank mean gene sets in **b**–**d**. Source data are provided as a Source Data file.

expression of osteoclast genes exhibited a correlative expression pattern of osteoblast genes (Fig. 2a). Thus, in the next set of analyses, we reasoned that the DMAb-suppressed osteoclast and osteoblast genes represented genes that may be biologically coupled even in the untreated (placebo) group. Consistent with this, plotting the rank means of DMAb-suppressed osteoclast and osteoblast genes in the placebo group demonstrated a clear, strong correlation of the coupling of osteoclasts and osteoblasts at the gene expression level (Fig. 2b, $R = 0.75$, $p = 0.002$; Spearman's rank correlation). Thus, using DMAb as a biological probe, we found that DMAb-suppressed osteoclast and osteoblast genes are biologically coupled in normal physiological conditions. In order to confirm the importance of these DMAb-suppressed osteoclast and osteoblast genes for coupling, we tested the correlation of these genes in a second independent dataset of whole, non-centrifuged bone from an entirely different cohort of participants, specifically untreated postmenopausal women ($n = 19$)[18]. *DCSTAMP* and *OCSTAMP* did not pass the limit of detection in this RNA-seq dataset, possibly because osteoclast genes could be underrepresented in the absence of centrifugation to remove marrow elements. However, similar to the centrifuged bone, the whole bone gene analysis showed a significant correlation between DMAb-suppressed osteoblast and osteoclast genes in untreated postmenopausal women (Supplementary Fig. 2), providing further confirmation of the validity of these findings.

**Correlations of gene sets with functional bone data**. To further validate the biological importance of the DMAb-suppressed osteoblast and osteoclast gene sets, we next correlated these

results with functional bone data from the placebo participants. We first correlated the serum bone remodeling measures in the placebo participants against the DMAb-suppressed osteoclast and osteoblast rank mean gene sets. Osteoblast genes downregulated by DMAb correlated positively with CTX and P1NP in the placebo participants (Supplementary Table 3A), showing a functional link between these osteoblast genes and systemic markers of bone remodeling. There was no correlation with serum TRAP5b or OCN. Using a segment of the bone biopsy preserved in paraformaldehyde prior to centrifugation, we next performed micro-computed tomography (μCT) to assess for bone geometric parameters in the placebo participant samples. Osteoclast genes downregulated by DMAb correlated negatively with Conn.Dens, Tb.N, and positively with Tb.Sp in the placebo participants, consistent with increased remodeling being associated with impaired bone microarchitecture (Supplementary Table 3B). In contrast to the serum markers of bone remodeling, correlation of osteoblast genes suppressed by DMAb did not reach significance with μCT parameters (BV/TV, Conn.Dens, SMI, Tb.N, Tb.Th, Tb.Sp). However, correlations trended in a similar direction to osteoclast genes downregulated by DMAb. Therefore, both DMAb-suppressed osteoblast and osteoclast gene sets correlate with functional bone parameters in postmenopausal placebo participants and further validate the use of DMAb as a biological probe to identify genes relevant for the coupling of bone resorption and bone formation in humans.

**DMAb-suppressed secreted factors correlate with gene sets**. Having validated the gene sets regulated by DMAb as biologically

relevant to osteoclast–osteoblast coupling, we next sought to identify potential secreted factors involved in this coupling in vivo in humans. For this, we used Ingenuity Pathway Analysis (IPA) to identify secreted factors significantly altered by DMAb in the centrifuged bone. We identified 55 suppressed secreted genes (Supplementary Fig. 3), and 51 upregulated secreted genes (Supplementary Fig. 4). Of interest, in the placebo participants, the rank means of DMAb-suppressed secreted factor genes significantly correlated with rank means for DMAb-suppressed osteoblast and osteoclast marker genes (Fig. 2c). In contrast, the DMAb-upregulated secreted factor genes did not correlate with either DMAb-suppressed osteoblast or osteoclast marker genes (Fig. 2d) in the placebo participants. These findings thus indicate that the DMAb-suppressed secreted factor genes may identify secreted factors necessary for coupling of osteoclasts and osteoblasts.

**Identification of osteoclast-derived coupling factors**. In order to identify which of the downregulated secreted factors may be osteoclast-derived, we assessed three additional human biopsy populations (Fig. 3a): osteocyte-enriched (serially digested bone fragments), osteoblast-enriched (alkaline phosphatase-positive (AP+) digest cells obtained through serial digest of flushed human bone biopsies), and bone marrow-derived osteoclasts. RNA-seq of the osteocyte-enriched fractions revealed that of the 55 secreted factors downregulated by DMAb in the centrifuged bone, 17 were also significantly decreased in the osteocyte-enriched bone (Fig. 3b). Three genes did not pass the limit for detection in osteocyte-enriched bone (*IMPG1*, *SERPINA12*, *SCGN*). This suggests that these three genes and the remaining 35 genes differentially regulated in the centrifuged bone, but not in the osteocyte-enriched bone, and may be genes that are differentially regulated in osteoclasts, osteoblasts, and bone lining cells, rather than osteocytes. Second, we compared gene expression of DMAb-suppressed secreted factors in bone marrow-derived osteoclast vs. osteoblast-enriched cultures by real-time quantitative PCR. Of the 55 secreted genes suppressed by DMAb, 10 genes were expressed at significantly higher levels in the osteoclast cultures compared to osteoblast-enriched fractions (Fig. 3c), including *LIF*, which has previously been identified as an osteoclast-derived coupling factor that stimulates bone formation in mice[19,20]. Taken together with factors differentially regulated in centrifuged bone vs. osteocyte-enriched bone, *LIF*, *CREG2*, *CST3*, *CCBE1*, and *DPP4* are most likely to be osteoclast-specific factors downregulated by DMAb, and to be potentially involved in coupling osteoclasts and bone resorption to bone formation in humans.

**Olink analysis of bone marrow plasma**. As a secondary approach to identify potential coupling factors linking osteoclasts and bone resorption to osteoblasts and bone formation, we utilized a high-multiplex proteomics approach (Olink Proteomics®; see Methods) to screen bone marrow plasma from placebo- and DMAb-treated women ($N = 24$/group). Forty-eight proteins were significantly altered in DMAb participant bone marrow plasma (Supplementary Table 4A). Of the osteoclast-derived secreted factors downregulated by DMAb at the RNA level, CST3 protein was significantly decreased in bone marrow plasma, and DPP4 protein levels trended lower (Fig. 4a, $p = 0.056$; Kruskal–Wallis). We next assessed correlation of the DMAb-suppressed osteoclast and osteoblast gene sets against the secretome data in the placebo patients for which we had RNA-seq and Olinks data (Supplementary Table 3B, $N = 15$/group). Of interest, DPP4 bone marrow protein positively correlated with osteoblast and osteoclast gene sets in placebo subjects (Fig. 4b, Supplementary Table 4B);

similar, although numerically smaller, correlations were found between peripheral serum DPP4 levels and the osteoblast and osteoclast gene sets in the placebo subjects (Supplementary Fig. 5). While the decrease in bone marrow plasma DPP4 using the Olink Proteomics® assay did not reach significance (Fig. 4a), analysis of the percent change in peripheral serum DPP4 levels revealed a significant decrease in serum DPP4 following treatment in the DMAb vs. placebo participants (Fig. 4c, $p = 0.008$; Mann–Whitney).

**Functional effect of reduced DPP4 in DMAb participants**. DPP4 is the target of the gliptin therapies for diabetes; inhibition of DPP4 prevents the proteolytic inactivation of glucagon-like peptide-1 (GLP-1)[21–23]. Thus, DPP4 inhibitors, such as sitagliptin, increase GLP-1 levels, leading to increased insulin synthesis and secretion, decreased glucagon release, and decreased plasma glucose. DPP4 has previously been shown to be increased in postmenopausal women and correlates with increased rates of bone turnover[24,25]. Since our mRNA expression analysis revealed DPP4 as an osteoclast-derived factor, we used in situ hybridization to assess for localization of *DPP4* mRNA expression in bone. Consistent with our mRNA expression approach, *DPP4* transcript was evident in osteoclasts on the bone surface, but not in lining cells, osteoblasts, or osteocytes in human cancellous and cortical bone (Fig. 4d). We next sought to determine if the reduction in DPP4 by DMAb (Fig. 4c) had a functional effect to increase GLP-1 levels and impact glucose homeostasis. Plasma taken before and after treatment revealed a significant increase in total GLP-1 in DMAb participants (Fig. 4e). However, DMAb did not significantly alter plasma glucose-dependent insulinotropic peptide (GIP) (Supplementary Table 5), glucose, or insulin levels in this healthy (non-diabetic) cohort of postmenopausal women (Fig. 4e). In addition, changes in blood lipids (total cholesterol, high-density lipoprotein (HDL) cholesterol, low-density lipoprotein (LDL) cholesterol, triglycerides) did not differ between the control and DMAb groups nor did changes in Homeostatic Model Assessment of Insulin Resistance (HOMA-IR) or HOMA-beta-cell function (HOMA-β) (Supplementary Table 5).

**DMAb improves glucose homeostasis in a diabetic cohort**. Because the noted lack of effect on glucose homeostasis may reflect a lack of impaired glucose metabolism (i.e., lack of metabolic syndrome or diabetes) in these healthy participants, we next assessed a group of diabetic patients treated clinically with DMAb vs. bisphosphonate or calcium plus vitamin D for 1 year. Baseline characteristics of the subjects are presented in Supplementary Table 6. By design, sex, body mass index (BMI), and type 2 diabetes or prediabetes duration did not differ among treatment groups. However, subjects in the DMAb group were slightly older than subjects in the other groups. In addition, more participants in the DMAb group were in the lifestyle-alone treatment for diabetes than in the other groups. Baseline hemoglobin A1c (HbA1c) and fasting plasma glucose (FPG) levels did not differ among groups.

Changes in HbA1c levels during the first 6 months and over the entire 12-month study period differed in the three groups ($p < 0.05$, $p < 0.0001$, respectively; analysis of covariance (ANCOVA) and Bonferonni). As shown in Fig. 5a, these differences were due to a greater decrease in HbA1c in the DMAb group relative to the bisphosphonate group in the first 6 months ($p < 0.05$) and a greater decrease in the DMAb group relative to both the bisphosphonate ($p < 0.00001$) and to the calcium plus vitamin D-alone group over 12 months ($p < 0.01$; ANCOVA and Bonferonni). Although the change in FPG did not differ among the three groups during the first 6 months or over

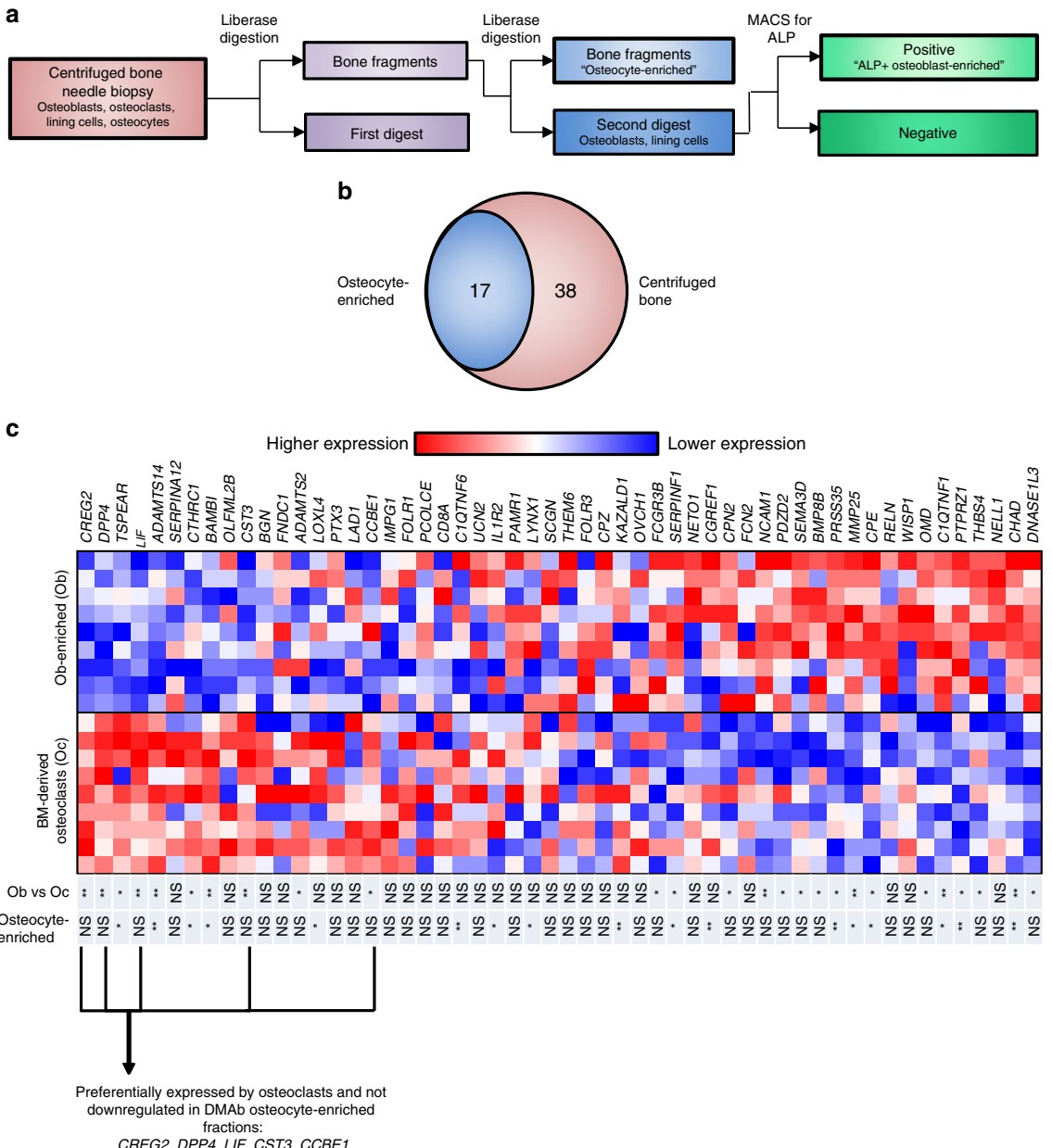

**Fig. 3 Identification of osteoclast-derived secreted factors involved in coupling. a** Flow chart describing processing of the bone biopsy samples to select for osteocyte- and osteoblast-enriched fractions. **b** Overlap of DMAb-suppressed secreted factor genes in centrifuged bone vs. the osteocyte-enriched fraction ($N = 15$ participant biopsies/group). For both sample sets, differential gene expression was determined using the R package edgeR. Ingenuity Pathway Analysis was used to identify differentially expressed secreted genes. **c** Heat map comparison of gene expression in osteoblast-enriched vs. bone marrow-derived osteoclast cultures. Red denotes higher expression and blue denotes lower expression. The Wilcoxon's signed-rank test was used to determine significance between osteoblast-enriched (Ob) and bone marrow-derived osteoclast (Oc) gene expression. *$P < 0.05$; **$p < 0.01$; NS not significant. Significance between osteoblast-enriched (Ob) and osteoclast (Oc) gene expression is shown in the Ob vs. Oc column ($N = 9$ participant samples/group). Significant change in osteocyte gene expression by DMAb compared to placebo is shown in the osteocyte-enriched column ($N = 15$ participant biopsies/group); differential expression of RNA-sequencing osteocyte data was performed with the R program edgeR. Source data are provided as a Source Data file.

12 months ($p = 0.07$, $p = 0.14$, respectively; ANCOVA and Bonferonni), the decrease was numerically greater in the DMAb group (Fig. 5b). The change in BMI differed among the three groups both over the first 6 months ($p < 0.05$) and over 12 months ($p < 0.001$, Fig. 5c; ANCOVA and Bonferonni). This was due to a greater decrease in BMI in the DMAb group than in the bisphosphonate group ($p < 0.01$) over the first 6 months and to a greater decrease in BMI in the DMAb group than in either the bisphosphonate group ($p < 0.0001$) or the calcium plus vitamin

D-alone group ($p < 0.01$; ANCOVA and Bonferonni) over 12 months. The change in BMI did not differ in the bisphosphonate group and the calcium and vitamin D-alone group over the first 6 months ($p = 0.09$) or over 12 months ($p = 0.16$; ANCOVA and Bonferonni). Importantly, the change in HbA1c levels in DMAb-treated patients remained significant after adjustment for changes in BMI (Supplementary Table 7). Thus, this clinical study demonstrated an improvement in glycemic control in type 2 diabetic or prediabetic patients with

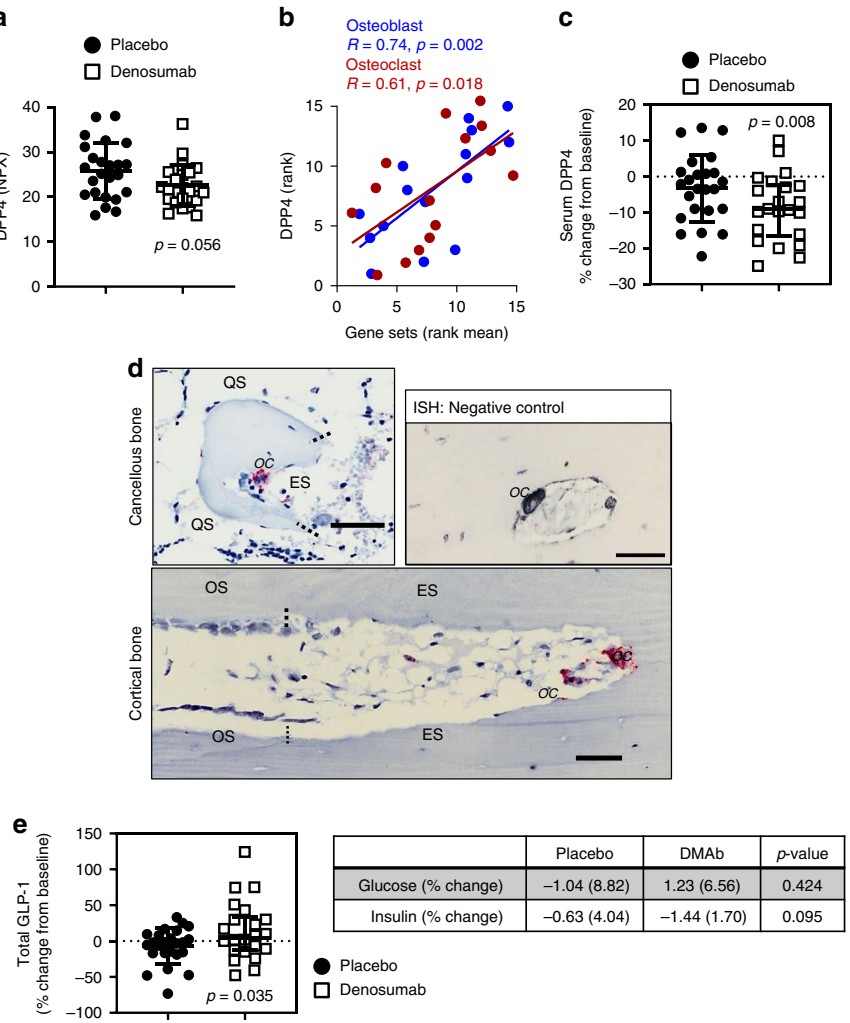

**Fig. 4 Bone marrow plasma DPP4 protein linked to bone remodeling. a** Bone marrow plasma DPP4 levels in the placebo- vs. DMAb-treated participants as measured by Olink Proteomics ($N = 24$ participant samples/group). Bone marrow DPP4 is measured in NPX (normalized protein eXpression), an arbitrary, relative unit in log 2 scale; values were converted to linear scale and $p$ value was calculated by the Kruskal–Wallis test. Individual values are plotted with mean and error bars represent SD. **b** Bone marrow plasma DPP4 levels (assessed by the Olink Proteomics) correlate with osteoblast and osteoclast gene sets in the placebo participants; Spearman's correlation coefficient was used to determine strength of relationships ($N = 15$ participant samples). **c** Change in serum DPP4 over 3 months in the placebo- vs. DMAb-treated participants (% change from baseline, $N = 24$ placebo, $N = 22$ DMAb participant samples [see Methods]); individual values are plotted with mean and error bars represent SD. Significance was determined using the Mann–Whitney test. **d** In situ hybridization (ISH) staining in human bone showing the presence of *DPP4* mRNA in osteoclasts, but not other cell types. Staining for *DPP4* mRNA (red stain) was abundant in osteoclasts (OC) on eroded surfaces (ES) of cancellous bone and intracortical canals. Osteoblasts on osteoid surfaces (OS) and bone lining cells on quiescent surfaces (QS) showed no staining. Dotted lines represent separation of bone surfaces. Scale bars = 50 μm. **e** Changes in serum GLP-1 (top) and glucose and insulin (bottom) levels in the placebo- and DMAb-treated participants (% change from baseline, $N = 24$ placebo, $N = 22$ DMAb participant samples). Individual values of percent change GLP-1 are plotted with mean and error bars represent SD. Significance was determined using the Mann–Whitney test. Source data are provided as a Source Data file.

osteoporosis who were treated with DMAb. Twelve months treatment with DMAb lowered HbA1c levels more than either a bisphosphonate or calcium plus vitamin D supplementation.

## Discussion

In this study, we utilized DMAb as a biological probe to pharmacologically ablate osteoclasts in postmenopausal women in order to identify potential secreted factors coupling bone resorption to bone formation. Using RNA-seq of centrifuged bone biopsies, we were able to identify DMAb-suppressed osteoblast and osteoclast gene signatures, and revealed the coupling of these sets at the gene expression level in untreated placebo participants. Importantly, this coupling of osteoblast and

osteoclast gene sets was confirmed in a second, independent sample of whole bone biopsies. Second, we identified secreted factors suppressed in centrifuged bone samples from DMAb-treated participants. Expression of these secreted factors strongly correlated with DMAb-suppressed osteoblast and osteoclast gene signatures in untreated placebo participants. By assessing additional cell populations from study participants, five of these genes were demonstrated to be osteoclast-derived secreted factors. Of significant interest, DPP4 was identified from this group as not only an osteoclast-derived protein with a possible role in osteoclast–osteoblast coupling but also a potential link between RANKL/bone remodeling and energy metabolism. Support for this conclusion was provided by our prospective finding that DMAb-treated participants had a significant reduction in

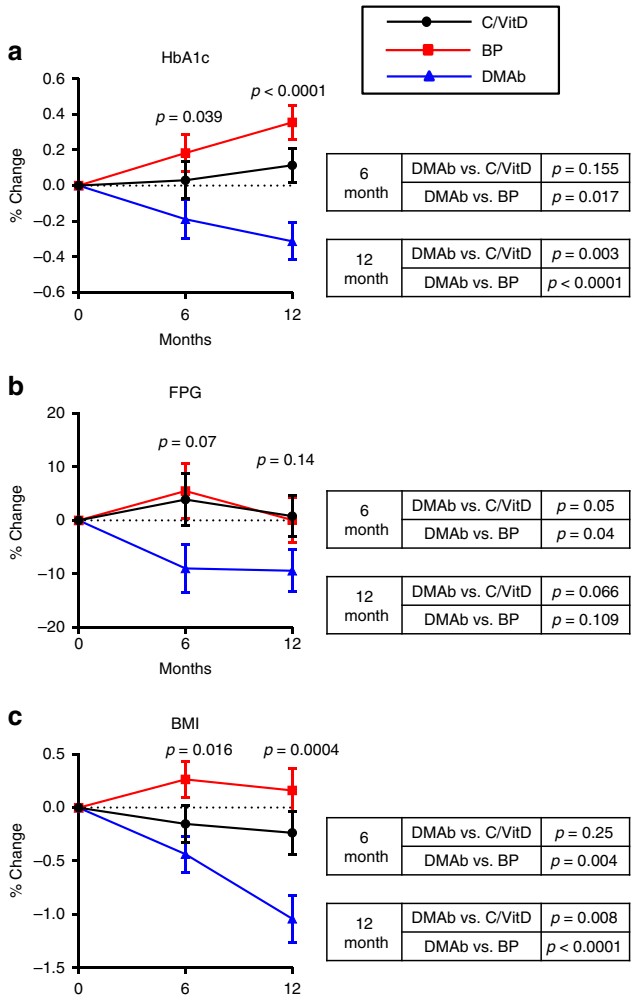

**Fig. 5 Analysis of type 2 diabetic/prediabetes patients treated with DMAb. a** HbA1c, **b** fasting plasma glucose (FPG), and **c** BMI are plotted for patients treated with calcium plus vitamin D (C/VitD, black solid line and circles), bisphosphonate (BP, red squares), or DMAb (blue triangles) ($N = 115$ patients per group). The y-axis shows percent change from baseline. Analysis of covariance (ANCOVA) was used to test for differences in continuous measures among the three groups. Bonferroni correction was used for post hoc pairwise comparisons. Mean percent change from baseline is presented at 6 and 12 months. Error bars represent SD. Source data are provided as a Source Data file.

circulating DPP4 levels and increase in GLP-1 levels as compared to the placebo-treated group, as well as a complementary analysis of type 2 diabetes patients, who were treated with DMAb vs. bisphosphonates or calcium plus vitamin D, in which DMAb-treated patients showed a significant improvement in HbA1c of a magnitude (relative to placebo or no treatment) comparable to commonly used anti-diabetic medications[26].

DMAb-suppressed secreted factors correlated strongly with both the osteoclast and osteoblast markers, whereas there was no correlation with DMAb-upregulated secreted factors. We used two strategies to identify factors potentially secreted by osteoclasts to stimulate bone formation. By assessing osteocyte-enriched fractions from placebo vs. DMAb participants, in which centrifuged bone was enzymatically stripped of lining cells such as osteoclasts and osteoblasts, we found that 17 of the genes were similarly reduced in osteocytes as compared to the centrifuged bone. This would then imply that the 38 secreted genes only

downregulated in centrifuged bone are regulated in cells lining the bone surface, but not in osteocytes. Secondly, comparison of AP+ digested cells from the bone cores to bone marrow-derived osteoclasts revealed secreted genes that were expressed at significantly higher levels in osteoclasts vs. osteoblasts. Of the genes with higher expression in osteoclasts that were suppressed in response to DMAb only in the centrifuged bone, we identified five genes: *LIF*, *CREG2*, *CST3*, *CCBE1*, and *DPP4*. Of interest, *LIF* has previously been identified as an osteoclast-derived factor, which promotes coupling of bone resorption to bone formation[19,20]. Thus, the identification of *LIF* through our independent analysis validates this approach to identify osteoclast-derived coupling factors. In addition, *CST3* has also been shown to induce osteoblast differentiation *in vitro*[27]. *CREG2* and *CCBE1* have yet to be assessed for potential roles in bone metabolism. Of these secreted factors, bone marrow concentrations of DPP4 and CST3 were found to be decreased in bone marrow plasma of DMAb-treated women ($p = 0.056$ and $p < 0.05$, respectively; Kruskal–Wallis), and serum DPP4 levels were significantly reduced following treatment with DMAb. Moreover, the bone marrow (and peripheral blood) concentrations of DPP4 correlated significantly with both osteoclast and osteoblast gene sets in placebo patients, consistent with previous reports that DPP4 correlates with increased bone remodeling in postmenopusal women[24,25].

DPP4 is a highly conserved peptidase found mostly on the surface of endothelial, epithelial, and immune cells[28]. It is found both on membranes as well as in soluble form. DPP4 has many substrates, with the best characterized targets being the incretin hormones, GIP, and GLP-1, both of which have anti-diabetogenic actions—most prominently stimulating β-cell secretion of insulin and suppression of glucagon secretion by α-cells. These actions are mediated in a glucose-dependent fashion[21–23]. Thus, DPP4 inhibitors have been developed and are clinically available for the treatment of type 2 diabetes. Soluble DPP4 levels have previously been demonstrated to be increased in postmenopausal women, consistent temporally with the higher rates of bone remodeling that occur following menopause[25]. In agreement with our findings, a previous report demonstrated expression of the membrane-bound form of DPP4, CD26, on osteoclasts in human bone in normal and pathological conditions[29]. As DPP4 is a peptidase targeting glycine/proline sequences, it is possible that osteoclast-derived DPP4 contributes to the degradation of collagen[30]. However, inhibition of the membrane-bound form of DPP4 appeared to inhibit osteoclast differentiation in vitro without affecting osteoclast resorptive activity[29]. Treatment of osteoblast cultures or mice with DPP4 inhibitors improved osteogenic differentiation and fracture healing[31]. In addition to likely direct effects of DPP4, the incretins targeted by DPP4 may have effects on both osteoblast[32–34] and osteoclast differentiation[35]. Other substrates for DPP4 that may be regulated within the bone microenvironment include stromal cell-derived factor-1, neuropeptide Y, peptide tyrosine tyrosine, insulin-like growth factor-1, vasoactive intestinal peptide, pituitary adenylate cyclase-activating polypeptide, and substance P[28]. Theoretically, the effects of the inactivation of various peptides might lead to differential effects on bone formation. While our findings are consistent with a correlation between DPP4 and rates of bone remodeling, questions still remain about DPP4 in relation to bone mineral density (BMD) and fracture incidence. Thus, the Cardiovascular Health Study found no association between serum DPP4 and BMD/fracture incidence[36]. In addition, Notsu et al.[37] also found no correlation between serum DPP4 levels and BMD in men with type 2 diabetes mellitus; however, a correlation of serum DPP4 with vertebral fracture incidence and bone turnover markers was present. Thus, further studies of DPP4 levels in relation to BMD and fracture incidence in healthy and diabetic

patients are needed to understand the contribution of DPP4 to bone metabolism.

Consistent with the important role of DPP4 in regulating incretins, the decrease in circulating DPP4 in our study was associated with a significant increase in total GLP-1 in DMAb vs. placebo participants. Although glucose and insulin levels were not affected in these participants, this is to be expected in a non-diabetic, healthy cohort, given the glucose dependency of GLP-1 actions on islet cell function. Similar results of a lack of an effect of DMAb on insulin resistance or lipid profiles in otherwise healthy, osteoporotic women was reported by Lasco et al.[38]. However, our analysis showing improved glycemic control in DMAb-treated diabetic patients indicates that the reduction in DPP4 levels and increase in GLP-1 levels do lead to a functional improvement in glucose homeostasis in individuals with abnormal glucose metabolism. Consistent with our findings, an analysis of the FREEDOM Trial did show a significant reduction in FPG in DMAb-treated women with diabetes or prediabetes not being pharmacologically treated for diabetes[39]. Nonetheless, we recognize that our case–control study in diabetic patients has inherent limitations, specifically lack of blinding and potential confounders, and a rigorous randomized controlled trial examining the effects of DMAb on glycemic control in patients with type 2 diabetes mellitus and osteoporosis is clearly warranted. In addition, we should note that we do not have data on the effect of DMAb on postprandial glucose metabolism, including insulin secretion and action. Evidence to date suggests that DPP4 inhibitors have effects on both fasting and postprandial glucose metabolism and that the effects are, in fact, more marked in the postprandial period[40]. Thus, a prospective study that measures postprandial β- and α-cell function together with glucose disposal is needed to further evaluate the mechanisms of the effects of DMAb on glucose homeostasis.

It will also be important to confirm our finding that DMAb treatment is associated with weight loss in a well-powered, randomized, placebo-controlled prospective study. That said, it is well established that DPP4 inhibitors may raise GLP-1 concentrations, but decrease net GLP-1 secretion[41–43]. This may limit the therapeutic efficacy of DPP4 inhibitors and prevent the degree of GLP-1 elevation necessary to modulate satiety and induce weight loss as observed with GLP-1 infusion or GLP-1 receptor agonist therapy[44]. Thus, it will be of interest to evaluate whether postprandial elevation of GLP-1 with DMAb exceeds that observed with DPP4 inhibitor treatment, thereby providing a potential explanation for the weight loss we noted following DMAb treatment.

Our finding that patients treated with DMAb had significant reductions in HbA1c levels as compared to bisphosphonate-treated patients suggests important differences between effects of these anti-resorptive drugs on glucose homeostasis. These differences may arise from differing effects of these two drugs on osteoclasts. Thus, although both drugs inhibit bone resorption, long-term bisphosphonate therapy is associated with little or no reduction in the number of osteoclasts, but with abnormal appearing osteoclasts that include giant, hypernucleated, detached osteoclasts[45]. By contrast, DMAb markedly reduces osteoclast numbers on all bone surfaces[46]. As such, bisphosphonates and DMAb may have differing effects on osteoclast DPP4 production, and this possibility warrants further investigation. In addition, as noted below, DMAb may have different effects on osteoblasts than bisphosphonates. Finally, DMAb likely modulates effects of RANKL on glucose homeostasis involving the pancreatic islets[47–49] or liver[50], thereby exerting more beneficial effects on glucose metabolism as compared to bisphosphonates.

Although we focused our analysis on osteoclast-derived secreted factors as candidate coupling factors, we should note

an important osteoblast-derived gene, *Wnt16*, that was significantly upregulated (2.23-fold, $P = 0.008$; Supplementary Table 2) in the DMAb relative to placebo bone biopsies. Previous work by the Ohlsson group has shown that osteoblast-derived Wnt16 inhibits osteoclastogenesis and prevents cortical bone fragility in mice[51], and SNPs in the *Wnt16* gene have been associated with cortical bone thickness[25] and fracture risk[52] in humans. The observed upregulation of Wnt16 expression by DMAb may provide an explanation for the finding from clinical trials that DMAb not only inhibits bone resorption, but enables continued bone mineral accrual following therapeutic intervention for up to 10 years of therapy[53]. This occurs even though markers of bone collagen synthesis remain suppressed, likely reflecting a consolidation of matrix mineralization enabled by DMAb, but not bisphosphonates.

Figure 6 provides a working model, based on our data, of the relationships between RANKL signaling in the osteoclast, release of

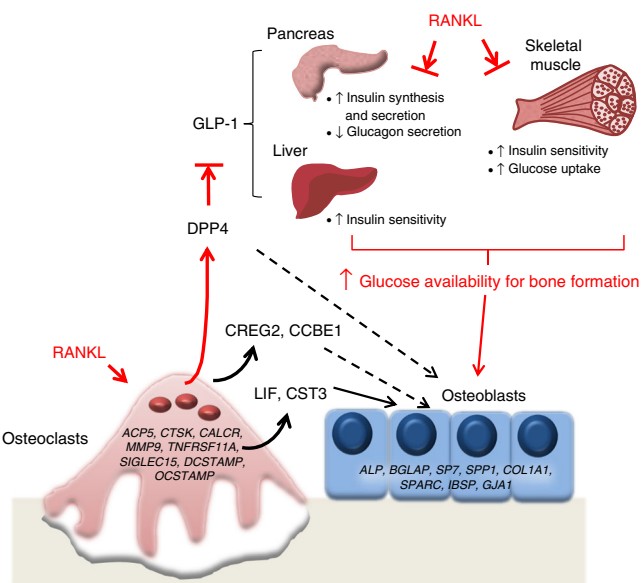

**Fig. 6 Proposed mechanism for coupling of bone remodeling to energy metabolism.** RANKL signaling in osteoclasts induces (directly or indirectly) the expression of key coupling factors, identified in our human study as *LIF*, *CREG2*, *CST3*, *CCBE1*, and *DPP4* (note that *LIF* and *CST3* are known to have effects on osteoblasts and thus are likely osteoclast–osteoblast-coupling factors (solid arrows)[19,20,27], whereas *DPP4*, *CREG2*, and *CCBE1* are potential coupling factors identified by our study that require further characterization (dashed arrows)). Increases in circulating DPP4 lead to a decrease in GLP-1 levels, leading in turn to reduced insulin and increased glucagon secretion, resulting in hyperglycemia. These actions of RANKL-induced DPP4 appear to be synergistic with other effects of RANKL (denoted by red arrows and text) on glucose metabolism, which are to induce insulin resistance and increase circulating glucose levels. These findings may be of particular interest in the context of previous findings from the Long laboratory demonstrating that Wnt signaling in osteoblasts favors glycolysis over oxidative phosphorylation (Warburg effect) and that glycolytic intermediates may be particularly important for the process of osteoblast differentiation[56]. As such, in addition to RANKL-inducing osteoclast differentiation and osteoclast-derived osteoblast-coupling factors, the ability to make glucose available to differentiating osteoblasts may also be a key component of RANKL-induced energy coupling at a systemic level. We should note that this is a working model and, for example, it remains to be shown that antagonism of RANKL by DMAb regulates the insulin/glucagon ratio in humans, not only under fasting but also postprandial conditions, given what is known about the mechanisms of action of DPP4 inhibitors[40].

osteoclast-derived coupling factors, including DPP4, and systemic glucose homeostasis. In this model, RANKL signaling in osteoclasts induces (directly or indirectly) the expression of key coupling factors, identified in our human study as *LIF*, *CREG2*, *CST3*, *CCBE1*, and *DPP4* (note that we can attribute induction of these factors to RANKL signaling as they were suppressed by DMAb treatment in our study). In addition to a potential role in coupling bone resorption to bone formation, DPP4 also has effects on systemic energy metabolism. Specifically, increases in circulating DPP4 would lead to a decrease in GLP-1, leading to reduced insulin and increased glucagon secretion, resulting in hyperglycemia. These actions of RANKL-induced DPP4 appear to be synergistic with other effects of RANKL on glucose metabolism, which are to induce insulin resistance and increase circulating glucose levels. Previous reports have shown that both liver and pancreatic β cells express the RANKL receptor TNFRSF11A. Activation of nuclear factor-κB-inducible kinase, downstream of RANKL, has been shown as a potential mechanism for obesity-induced β-cell failure[47]. In addition, inhibition of RANKL via DMAb or osteoprotegerin (OPG) stimulates β-cell proliferation[48]. In the liver, it was found that hepatocyte-specific knockout of TNFRSF11A protected against insulin resistance in mice[50]. Finally, OPG-knockout mice show a significant increase in fasting glucose[49]. These previously defined actions of RANKL on inducing insulin resistance may be linked to our finding that DMAb reduced circulating DPP4 and increased GLP-1 levels, in that both DPP4 inhibitors[54] and GLP-1[55] may also reduce insulin resistance. Thus, considerable evidence now indicates that the net effect of RANKL signaling is to impair insulin secretion, induce insulin resistance, and increase circulating glucose levels (Fig. 6). These findings may be of particular interest in the context of previous findings demonstrating that Wnt signaling in osteoblasts favors glycolysis over oxidative phosphorylation (Warburg effect) and that glycolytic intermediates may be particularly important for the process of osteoblast differentiation[56]. As such, in addition to RANKL-inducing osteoclast differentiation and osteoclast-derived osteoblast-coupling factors, the ability to make glucose available to differentiating osteoblasts may also be a key component of RANKL-induced energy coupling at a systemic level. Indeed, consistent with this hypothesis, Bonnet et al.[57] recently demonstrated that transgenic mice overexpressing RANKL have systemic insulin resistance and reduced glucose uptake in non-skeletal tissues (muscle, brain), but increased glucose update in bone (femur). Although evident in the adult, as in our study, these effects of RANKL may be particularly important during rapid growth and development, when the energy needs of the skeleton are substantial. Overall, our data indicate that RANKL-induced DPP4 may be part of this process linking bone to energy metabolism.

In summary, our study is the first to independently verify LIF as a coupling factor in human bone remodeling, identify several additional coupling factors in humans, and uncover osteoclast-derived DPP4 as a potential link between bone remodeling and energy metabolism. Our finding that DMAb suppression of DPP4 may contribute to improved glucose homeostasis in patients with type 2 diabetes also provides a strong rationale for future studies examining the role of RANKL inhibition in simultaneously treating osteoporosis and diabetes, two common age-related co-morbidities.

## Methods

**Study subjects.** We used data from two cohorts: a prospective, randomized controlled study where participants were treated either with placebo or denosumab (study A) and a clinical cohort where patients with type 2 diabetes were assigned by their physicians to treatment with calcium/vitamin D, a bisphosphonate, or denosumab (study B).

*Study A*: This was a clinical trial by the NIH definition, and was registered in ClinicalTrials.gov (Identifier NCT02554695). Additional details are provided below:

*Pre-specified outcomes.* Gene expression changes in bone cells between groups, specifically genes related to osteoclast–osteoblast-coupling factors.

*Inclusion/exclusion criteria.* All women were aged 50–80 years old with a BMI between 18 and 34 kg/m², ≥5 years since last menses, and rigorously screened for coexisting disease. Exclusion criteria were as follows: (1) abnormality in any of the screening laboratory studies (complete blood count, serum calcium, phosphorus, albumin, AP, creatinine, aspartate aminotransferase, 25-hydroxyvitamin D (25 (OH)D), and thyroid-stimulating hormone); (2) any fracture within the previous 6 months; (3) presence of stage IV or V chronic kidney disease, chronic liver disease, severe neuropathic disease, unstable cardiovascular disease, malignancy, chronic gastrointestinal disease, hypo- or hyperparathyroidism, Cushing's syndrome, severe chronic obstructive pulmonary disease, alcoholism, or type 1 diabetes; (4) undergoing treatment for blood clots, coagulation defects, or treatment with any of the following drugs: corticosteroids, anticonvulsant therapy, pharmacological doses of thyroid hormone, adrenal or anabolic steroids, aromatase inhibitors, calcitonin, calcium supplementation >1200 mg/day, bisphosphonates, estrogen, selective estrogen receptor modulators, parathyroid hormone, sodium fluoride, teriparatide, or thiazolidenediones. All subjects were required to have sufficient levels of vitamin D (serum 25(OH)D of >20 ng/mL).

*Approval procedure by IRB.* The protocol was reviewed first by the Department of Endocrinology Scientific Review Committee and then reviewed and approved by the Mayo Institutional Review Board (IRB) (Mayo IRB #15-002313, "Effects of age and osteoclast inhibition on bone formation."). All participants provided written, informed consent prior to participation in the study.

Because the study was originally designed to examine osteoclast–osteoblast-coupling factors, we did not include fasting glucose as a screen. When we subsequently analyzed the glucose values in the prospective study, we discovered that 1 of the 48 subjects did, in fact, have type 2 diabetes. Because clinical trials and studies in osteoporosis do not generally exclude patients with type 2 diabetes, we continued to include this subject in the gene expression analyses, although we would note that excluding this subject did not change the results of that analysis. However, for the analyses specific to glucose metabolism (Fig. 4c, e, Supplementary Table 5), we excluded this subject due to her abnormal glucose values, and restricted the glucose homeostasis analysis to a non-diabetic cohort. This allows for a cleaner analysis of this cohort, and the possible effects of denosumab in patients with type 2 diabetes is specifically addressed in the subsequent clinical study (see below).

In calculating the HOMA-IR and HOMA-B parameters, we discovered that a second subject had, contrary to our instructions, clearly eaten before her blood draw (high glucose, insulin, and GIP values that were elevated at baseline but normal at her 3-month visit). Because this would alter all of the parameters related to changes in glucose homeostasis with denosumab (but not her gene expression data, as that was obtained at the 3-month visit, when she did fast), we have also excluded her from the glucose homeostasis analysis in this cohort (Fig. 4c, e, Supplementary Table 5).

*Study B*: This was not a clinical trial by the NIH definition, as it did not prospectively recruit and randomize subjects to treatment. Rather, the patients were being treated clinically by their physicians, and we evaluated changes in glycemic parameters in response to the clinically assigned treatment through the electronic medical record. As such, this study was not registered in a clinical trial database.

*Pre-specified outcomes.* Based on data in the electronic medical record, the subjects' FPG and HbA1c at baseline, 6 months, and 12 months of treatment.

*Inclusion/exclusion criteria.* Subject population: Patients with type 2 diabetes mellitus and osteoporosis treated at Mayo Clinic Rochester.

Inclusion criteria. Patients aged 45–100 years old; patients diagnosed with type 2 diabetes mellitus/prediabetes and osteoporosis; treated with denosumab, bisphosphonate, or calcium/vitamin D alone for osteoporosis, for at least 1 year.

Exclusion criteria. Age <45 years or >100 years old; type 1 diabetes mellitus or lack of prediabetes; treated with denosumab or bisphosphonate for <1 year.

*Approval procedure by IRB.* The protocol was reviewed first by the Department of Endocrinology Scientific Review Committee and then reviewed and approved by the Mayo IRB (Mayo IRB #16-001907, "Glycemic control of type 2 diabetic or prediabetic patients with osteoporosis treated with denosumab vs. bisphosphonate."). As this study was conducted through the electronic medical record without active recruitment, written informed consent was not required by the Mayo Clinic IRB. However, all patients had provided authorization to Mayo Clinic for use of their medical records for research.

**Study A protocol.** Following recruitment, subjects were randomized into two groups: placebo and DMAb. At baseline, morning fasting blood was drawn between 7 and 8 a.m. Weight was obtained using an electronic scale and height was measured using a customized stadiometer. Placebo subjects received a saline subcutaneous injection, whereas the subjects randomized to DMAb received a single subcutaneous injection of DMAb (60 mg). Three months following the treatment, subjects returned to the Mayo Clinical Research Trials Unit; the baseline blood

draw was repeated, and the subjects underwent sampling for bone marrow plasma and needle bone biopsies.

**Obtaining and processing needle biopsies of bone.** We obtained four small needle bone biopsies from the posterior iliac crest of all subjects using an 8 G needle under local anesthesia (1% lidocaine) and monitored intravenous (IV) sedation (1–3 mg of IV midazolam and 50–100 μg of fentanyl), as described previously[17]. All biopsies were performed without complications. Each bone biopsy was 1–2 mm in width and 1–2 cm in length, and contains both cortical and trabecular bone. A small section of sample #1 was preserved in 4% paraformaldehyde for further analysis. The remainder of sample #1 was centrifuged at max speed for 30 s to remove loosely adherent marrow. The remaining bone with lining cells was immediately homogenized (Tissue Tearor, Cole-Parmer) in QIAzol (Qiagen, Valencia, CA) for RNA isolation and stored at −80 °C until further processing. Samples #2–4 were combined and digested;[17] the bone was minced using a scalpel and was incubated for 30 min, 37 °C with Liberase (Roche) according to the manufacturer's protocol. Following the first digest, bone was washed three times with phosphate-buffered saline (PBS); remaining bone was re-minced with a scalpel and subjected to a second round of Liberase digestion, followed by PBS washes. The washes were combined with cells collected in the second digest. The stripped bone fragments from samples are enriched for osteocytes; these samples were homogenized in QIAzol and stored at −80 °C prior to RNA isolation. The cells collected from the second digest (and washes) were incubated with biotinylated antibody to human AP as per the manufacturer recommended concentrations (BAM1448, R&D Systems, Minneapolis, MN), followed by incubation with anti-biotin magnetic beads (Miltenyi Biotec) and sorted by magnetic activated cell sorting to obtain cells positive for AP (AP+ osteoblast enriched)[17]. AP+ cells were immediately lysed for subsequent RNA isolation (RNeasy, Qiagen).

**RNA isolation and gene expression analyses.** Total RNA was isolated using the RNeasy Mini Kit (Qiagen, Valencia, CA) and treated with on column DNase to remove potential contaminating DNA. RNA from centrifuged bone and osteocyte-enriched fragments were submitted for RNA-sequencing. Library preparation and sequencing were performed as described previously[17]. Briefly, first-strand complementary DNA (cDNA) was generated from ~100 ng of total RNA using DNA/RNA chimeric primers and reverse transcriptase, creating a cDNA/RNA hybrid, followed by second-strand cDNA synthesis containing a DNA/RNA duplex. The resulting double-stranded cDNA products were modified by random priming and extension to create double-stranded products suitable for generating sequencing libraries. The double-stranded products then underwent blunt-end repair. Adapter molecules were ligated to the 5′ and 3′ ends of each fragment to facilitate PCR amplification of the fragments to produce the final library. Unique indices were created for each sample and incorporated at the adaptor ligation step for loading multiple samples per flow cell. Libraries were sequenced on an Illumina HiSeq 2000 using TruSeq SBS sequencing software (version 3) and SCS data collection software (version 1.4.8). Base calling was performed using Illumina RTA (version 1.12.4.2).

RNA isolated from AP+ osteoblast-enriched digest cells and bone marrow-derived osteoclast cultures were amplified to cDNA using the Ovation RNA Amplification System V2 (NuGEN Technologies Inc., Redwood City, CA). Amplified cDNA was then used to perform quantitative real-time PCR (QPCR) for secreted genes downregulated by DMAb in centrifuged bone, as well as a panel of housekeeping genes (Supplementary Table 7). QPCR reactions were run using the ABI Prism 7900HT Real-time System (Applied Biosystems) with RT2 SYBR Green Mastermix (Qiagen, Valencia, CA). Normalization for variations in input RNA was performed based on the panel of housekeeping genes (*ACTB*, *B2M*, *G6PD*, *GAPDH*, *GUSB*, *POLR2A*, *RPL13A*, *TBP*, *TUBA1A*) from which the three most stable housekeeping genes were selected using the geNorm algorithm (http://medgen.ugent.be/~jvdesomp/genorm/)[58,59].

**Serum markers of bone metabolism.** Serum markers of bone metabolism were measured using the following assays: serum CTX—One-step ELISA Kit (Nordic Bioscience Diagnostics, coefficient of variation [CV] <8%); PINP—RIA (Orion Diagnostica, CV <9%); TRAP5b—ELISA (Immunodiagnostic Systems Ltd., CV <14%); OCN—ELSA-Osteo Two-site immunoradiometric assay (Cisbio US, CV <8%); 25(OH)D (CV <7%)—liquid chromatography-tandem mass spectrometry (API 5000; Applied Biosystems/MDS SCIEX).

**Osteoclast differentiation.** Patient bone marrow was diluted 7:1 in wash buffer (sterile PBS, pH = 7.2, 2 mM EDTA) and layered onto Ficoll-Paque as instructed by the MACS Miltenyi Biotec Protocol; bone marrow was centrifuged for 35 min at 400 × g with no brake, allowing for separation of mononuclear cells. Mononuclear cells from the interphase were collected, washed in wash buffer, and counted for plating of osteoclast differentiation assays. Cells were differentiated in α-MEM (Gibco) with 10% fetal bovine serum (Hyclone) and 1× antibiotic/antimycotic, and 35 ng/mL human recombinant macrophage colony-stimulating factor (Peprotech) and 100 ng/mL human recombinant RANKL (Peprotech). Differentiation media were changed every 3 days, and osteoclasts were detected after 7 days of differentiation. Cells were lysed with Buffer RLT (Rneasy, Qiagen) and used for subsequent RNA isolation as described above.

**Olinks proteomics.** An aliquot of bone marrow plasma from each subject was submitted on a 96-well PCR plate to Olink Proteomics. Samples were run on the following panels: Cardiometabolic, Cardiovascular II, Cardiovascular III, Immuno-Oncology, and Inflammation. The assays are designed around Precision Extension Assay technology in which each biomarker is targeted by a pair of antibodies coupled to partially complementary oligonucleotides, and measurements of each biomarker is made via real-time PCR. Data are provided as Normalized Protein eXpression (NPX), an arbitrary unit in log 2 scale, calculated from Ct values and pre-processing normalization; biomarker NPX values are relative units.

**In situ hybridization.** Formalin-fixed, decalcified, and paraffin-embedded bone specimens from five human controls were included for in situ hybridization analysis. Four of the human bone specimens were diagnostic iliac crest biopsies obtained from adult control patients formerly under investigation for a hematological disorder, as previously described[60]. One of the human bone specimens was obtained from the proximal femur of an adolescent patient during corrective surgery for coxa valga, as previously described[61]. The study was approved by the Danish National Committee on Biomedical Research Ethics (S-20070121 and S-20120193). Paraffin sections (3.5-μm-thick) were subjected to in situ hybridization using an enhanced version of the RNAScope 2.5 high definition procedure (310035, ACD Bioscience). Sections were rehydrated, deparaffinized, and pretreated as previously[60], and hybridized overnight at 40 °C with 20-ZZ-pair probes (477541, ACD Bioscience) directed against the 2329–3361 region of human *DPP4* mRNA (NM_001935) diluted 1:1 in probe diluent (449819, ACD Bioscience). A negative control only hybridized with probe diluent was included. The amplification was conducted according to the instructions provided by the manufacturer. The horse radish peroxidase was further enhanced with digoxigenin (DIG)-labeled tyramide (NEL748001KT, PerkinElmer), which was labeled with AP-conjugated sheep anti-DIG FAB fragments (11093274910, Roche) and visualized with Liquid Permanent Red (Dako, Denmark). Finally, the sections were counterstained with Mayer's hematoxylin and mounted with Aqua-Mount.

**Serum/plasma measures of glucose and lipid homeostasis.** Plasma glucose was measured on the Roche Cobas c311 (Roche Diagnostics, CV <1.4%). Serum DPP4 was measured using an ELISA Kit (R & D Systems, CV <8.4%). Plasma insulin was measured by two-site immunoenzymatic assay on the Dxl 800 Automated Immunoassay System (Beckman Instruments, CV <7.7%). Plasma GIP was measured by a quantitative ELISA (Millipore Corp., CV <7.2%). Total GLP-1 was measured using a direct, double antibody RIA (Millipore Research, CV <13%). Total cholesterol, HDL cholesterol, and triglycerides were measured by enzymatic colorimetric assays on the Roche Cobas c311 analyzer (Roche Diagnostics, CVs <2.6%).

**Study B protocol.** This was a case–control study of subjects with type 2 diabetes mellitus or prediabetes with osteoporosis treated at Mayo Clinic in Rochester, Minnesota, Scottsdale, Arizona, and Jacksonville, Florida, between 1 January 2009 and 13 July 2016. The starting date was selected because DMAb was approved by the Food Drug Administration in 2009. The subjects were divided into three groups based on treatment of their osteoporosis with DMAb, oral or intravenous bisphosphonate, or calcium and vitamin D supplementation without anti-resorptive medication. FPG, HbA1c, and weight were assessed at baseline, 6 months, and 12 months after subjects started osteoporosis treatment.

**Study participants.** Eligible participants were between 45 and 100 years old, diagnosed with osteoporosis and type 2 diabetes mellitus or prediabetes, and treated with DMab, oral or intravenous bisphosphonate, or calcium and vitamin D supplementation for at least 1 year. After approval by the Mayo Clinic IRB, all subjects were identified with the i2b2 search engine in the Mayo Clinic patient clinical database, and their medical records were reviewed. Subjects were excluded if baseline, or 6 or 12 months HbA1c, FPG and weight, and duration of type 2 diabetes mellitus or prediabetes were not available. A total of 253 potential subjects with type 2 diabetes mellitus or prediabetes treated with DMab for osteoporosis were identified. Of these, 138 were excluded due to failure to meet inclusion criteria, and 115 were eligible for inclusion in the DMab-treated group. One hundred and fifteen subjects treated with a bisphosphonate (either oral: $n = 83$ or intravenous: $n = 32$) and 115 subjects treated with calcium and vitamin D who matched the DMab subjects for age, sex, BMI, and duration of type 2 diabetes or prediabetes also were identified.

**Study end points.** The primary end point of this study was the change in HbA1c from baseline to 6 months and from baseline to 12 months after starting osteoporosis treatment. The secondary end points were the changes in FPG and body weight from baseline to 6 months and from baseline to 12 months after starting osteoporosis treatment.

**Statistical analysis.** The RNA-seq data was processed using Mayo Clinic's MAP-Rseq (v2.1.0) application[62], which is a computational pipeline for the analysis of Illumina's paired end RNA-seq reads. MAP-RSeq uses a mix of locally developed

methods and publically available bioinformatics tools. Within MAP-RSeq, TopHat2[63] with the bowtie1 option was called to align each sample's reads to the hg19 reference genome. The first 100,000 reads of each sample was used to estimate the mean and the standard deviation of the fragment length, which is required information for TopHat. The gene counts were generated by FeatureCounts[64] using Ensembl's hg19 gene definition file. The "-O" option within FeatureCounts was used to account for the expression derived from regions shared by multiple genomic features. RSeqQC[65] was used to create quality control metrics, including gene body coverage plots, to insure the results from each sample were reliable and could be collectively used for a differential expression analysis.

The R package edgeR[66] was used to identify which genes were differentially expressed between the Placebo and DMAb-treated subjects. Technical variability introduced by gene GC content, gene size, and total calls per sample was accounted for using an offset obtained from the cqn package[67]. Genes with a median gene count of <10 in all the groups were excluded. Relationships between the osteoblast and osteoclast genes involved taking all the 28,402 genes for a given subject and ranking their CQN normalized values from high to low, then taking the mean values of the ranks for the osteoblast genes and for the osteoclast genes. The Spearman's correlation coefficient was used to summarize the relationship between the mean osteoblast and osteoclast ranks. A randomization test was used to assess the significance of this correlation, using randomly created sets of genes and assessing the correlation between them. This process was carried out 5000 times and the correlation of the osteoblasts/osteoclasts was compared to these other 5000 correlations. Spearman's correlation coefficients were also used for comparisons between the mean rank values and serum biomarker values. The Kruskal–Wallis test was used to assess group differences for the Olink data.

For the cohort consisting of diabetics, prediabetics, and controls, ANCOVA was used to test for differences at baseline and change from baseline in continuous measures amongst the three groups. Sex and baseline diabetes treatment were compared using a $\chi^2$ test. HbA1c and FPG ANCOVA models were adjusted for age, BMI, sex, type 2 diabetes or prediabetes duration, and diabetes treatment. Weight models were adjusted for age, sex, type 2 diabetes or prediabetes duration, and diabetes treatment. Longitudinal mixed models were used to test for overall differences across treatment groups during all time points for HbA1c and FPG and weight. Predicted marginal mean values for the different groups were calculated using the overall distribution of the adjustment variables in the study population. A $p$ value of <0.05 was considered statistically significant for overall three groups comparisons, and a Bonferroni-corrected threshold of $p < 0.017$ was used for post hoc pairwise comparisons. Statistical analyses for all analyses were conducted using SAS (version 9.4; Cary, NC) and R (version 3.4.2; Vienna, Austria). All tests were two sided.

**Reporting summary**. Further information on research design is available in the Nature Research Reporting Summary linked to this article.

## Data availability

The data that support the findings of this study are available from the corresponding authors upon reasonable request. Source data underlying all Figures and Tables are provided as a Source Data file. Differential gene expression analysis generated from RNA-seq of centrifuged bone biopsies is included in the Source Data file, and individual raw data is available at the NCBI's Gene Expression Omnibus (GEO) site as follows: RNA-seq data from centrifuged (GSE141610) and osteocyte-enriched (GSE141595) bone biopsy samples: https://www.ncbi.nlm.nih.gov/geo/query/acc.cgi?acc=; GSE141614 RNA-seq data from whole bone biopsy samples:[18] https://www.ncbi.nlm.nih.gov/geo/query/acc.cgi?acc=; GSE72815 Olink proteomic datasets: The data files for the Olink proteomic analysis are attached as a supplementary file in Excel (Olink NPX raw data_Weivoda et al.)

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

## Acknowledgements

This work was supported by NIH grants AG004875 (S.K., D.G.M.), AR067129 (M.J.O., S.K.), AR027065 (S.K.), AR070281 (M.M.W.), and AR068275 (D.G.M.); and James A. Ruppe Career Development Award in Endocrinology (M.M.W.). We also wish to acknowledge the infrastructure support of the Mayo CTSA (UL1 TR002377).

## Author contributions

S.K. and M.J.O. conceived the idea for study A. Study A recruitment was performed by A.J.T. and L.K.M. Biopsy collections were carried out by L.K.M. and M.T.D. Samples were processed and cell isolations were carried out by M.M.W., D.G.M., J.N.F., J.R.G., B.E., B.T., and M.R. RNA isolations and cDNA library prep were performed by B.E., D.G.M., and M.R. ELISAs of bone serum markers was performed by B.T. RNA-seq data was analyzed for differential expression by E.J.A. Analysis of differentially expressed genes was performed by M.M.W. leading to the observation of gene set correlations. After discussion with S.K., M.J.O., and D.G.M., further correlation analysis was initiated by M.M.W. Validation of statistical significance of correlations across gene sets was performed by M.M.W. and E.J.A. Analysis of Olink proteomic data was performed by M.M.W. and E.J.A. Statistical analysis of metabolic serum parameters was carried out by E.J.A. In situ staining was performed by T.L.A. and M.K. Discussion of data was performed with M.M.W., D.G.M., J.N.F., R.A.R., A.V., A.M., B.L.C., M.J.O., and S.K. Study B was performed by C.K.C. and B.L.C. Statistical analysis for Study B was performed by E.J.A. The manuscript was written by M.M.W., C.K.C., and S.K. with input from all authors, and revised by M.M.W. and S.K. with input from all authors.

## Competing interests

The authors declare no competing interests.
