## [Peer Review File · Nature Communications]

Reviewers' Comments:

Reviewer #1:

Remarks to the Author:

The skeleton is a metabolically active endocrine organ. The best evidence for this comes from studies of FGF23, an osteocyte-derived phosphaturic hormone that also regulates renal 1-alpha hydroxylase activity, parathyroid hormone production, and cardiomyocyte hypertrophy. Preclinical murine models have suggested that the skeleton controls fuel metabolism via osteocalcin-regulated insulin and adiponectin production. However, any evidence for this or a similar bone-pancreatic endocrine axis in humans has been entirely unconvincing.

In the present study, Oursler, Khosla and colleagues provide the very first robust evidence of a skeletal - pancreatic endocrine axis that regulates fuel metabolism in humans. RANKL, an osteocyte-derived paracrine signal that promotes osteoclast-mediated bone loss, is inhibited by denosumab (D-mab) as an effective therapy for osteoporosis. During their studies deploying RNA-Seq, interrogating the transcriptome in bone from patients treated with D-Mab vs. placebo, the investigative team identified that skeletal expression of DPP4, a proteolytic inhibitor of the incretin GLP1, was down-regulated with D-Mab therapy. In situ hybridization identified that DPP4 was expressed in the osteoclast lineage. Consistent with this biology, bone marrow plasma and circulating levels of serum DPP4 were reduced in cohorts of patients treated with D-Mab ($p=0.02$) with concomitant increase in baseline circulating GLP1 ($p = 0.05$). Because there was no significant impact of D-Mab on glucose or insulin homeostasis in otherwise healthy (non-diabetic) women with osteoporosis, the investigative team studied a case-control cohort of patients with pre-diabetes or frank T2D treated with D-Mab vs. an aminobisphosphonate (aminoBP) vs. calcium+vitamin D for 1 year. Unlike D-Mab therapy, wherein osteoclast numbers are profoundly reduced (e.g. a pure "clastocide"), aminoBP therapy results in reduced osteoclast resorption activity in the setting of modest skeletal accrual of multinucleated osteoclast numbers. As compared to either aminoB or calcium plus vitamin D, HbA1c levels reduced in this setting, with concomitant reductions in fasting plasma glucose. BMI, and important contributor to insulin resistance in the setting of MetS and T2D, is also reduced by D-mab. Thus, the authors conclude that osteoclast-derived DPP4 as a RANKL-ligand regulated link between bone remodeling and fuel metabolism --thereby providing the very first robust evidence of a skeletal - pancreatic endocrine axis that regulates fuel metabolism in humans as noted above. I have only the following suggestions concerning this very important and enlightening manuscript:

1 — Lipid profiles, particularly serum triglycerides, are important contributors to the cardiometabolic risk profile of MetS-T2D patients, often entrained to insulin sensitivity. Have the authors identified whether fasting triglycerides are reduced in D-mab treated cases vs. cohort controls (Ca+D, aminoBP)? Alternatively, were plasma HDL levels, often reciprocally related to serum Trigs, increased with D-Mab? Any such data would be a welcome addition, and provide additional evidence for their working model in Figure 6.

2 - Page 19, Methods: Because GLP1 varies so dramatically based upon prandial setting, etc., it would be very helpful to describe the conditions under which serum was drawn for studies of GLP1 and DPP4 in their human subjects. Were these drawn under fasting conditions?

3- If the authors have calculated the HOMA-IR (Homeostatic model assessment of insulin resistance) and/or HOMA-beta for their relevant cohorts, this could be included in the online supplement (see also comment 5 below).

4 -- The authors may wish to briefly discuss the limitations of their case-cohort study design for the benefit of the Journal's broad audience.

5 - In their legend to Figure 6, the authors should mention that it remains to be shown that

antagonism of RANKL by D-Mab regulates the insulin / glucagon ratio in humans. If, however, the authors have calculated the HOMA-IRs / HOMA-betas for their cohorts this could be included.

6— Unlike aminoBPs, the RANKL ligand antagonist Denosumab (D-mab) not only inhibits resorption but enables continued bone mineral accrual following therapeutic intervention beyond 2 years of therapy (plateaus on aminoBPs). This occurs even though markers of bone collagen synthesis remain reduced, a consolidation of matrix mineralization enabled by D-Mab but not aminoBPs. The molecular mechanisms contributing to this important biology that favorably “uncouples” bone mineral accrual from bone resorption is completely unknown. The authors data identified that skeletal expression of Wnt16, an important contributor to human (and mouse) bone mass as determined genetically, is increased with D-Mab (data supplement). Indeed, this discovery may be as important as the identification of the novel coupling factors emphasized. The authors may wish to very briefly comment on this observation in their discussion as well.

Minor comments:

7- The observation that D-mab increases GLP1 while reducing DPP4 and BMI is intriguing. While GLP1 receptor agonists also cause weight loss, DPP4 antagonists are usually weight neutral in T2D. Do the authors have any thoughts as to the mechanisms of BMI reduction with D-mab therapy?

8- Page 17, magnetic misspelled as “magentic.”

Reviewer #2:

Remarks to the Author:

In the study by Weivoda et al. a single DMAB treatment was given as a biological probe to ablate osteoclasts in healthy postmenopausal women to identify potential secreted factors coupling bone resorption and bone formation. Based on RNA-sequencing of 15 bone samples per group they identified LIF, CREG2, CST3, CCBE1 and DPP4 as possible osteoclast-derived factors which are downregulated by DMAB, and might be involved in the coupling of bone resorption to bone formation. Circulating DPP4 was significantly reduced in DMAB treated participants when compared to the placebo-treated group. In addition, using a prospective study, DMAB treatment reduced HbA1c levels in type 2 diabetic patients as compared to patients either treated with bisphosphonates or calcium and vitamin D. Thus, this study highlights new coupling factors of bone resorption and formation in humans.

This study appears extremely well done in terms of obtaining high-quality RNA-seq data on bone and the collection of prospective cohort data. The data they have obtained in terms of bone remodeling genes is convincing and this kind of cohort is unique. Even though this is a beautiful proof-of-concept study, many of the results confirm what is already known and novel mechanistic insights are limited, even though novel candidates are proposed.

One exciting aspect is the more intricate analysis of DPP4. Bone marrow levels of DPP4 correlated with bone turnover gene sets. Was this also the case for serum DPP4 levels? What about GIP? As this is also a target of DPP4, was this parameter also assessed in the cohort and does it correlate with bone turnover? Furthermore, were DPP4 and GLP-1 levels also measured in the diabetic cohort? Did this correspond with HbA1c and/or bone turnover markers? In general, it would be interesting to show BMD (T-score) and bone turnover makers in the diabetic cohort and correlate these with the DPP4 targets, possibly even with HbA1c and FPG.

Concerning the diabetic cohort, more patients in the DMAB group were on life style intervention. Was that the reason for the more significant reduction of BMI as compared to the other groups? May this be a confounder for the analysis of diabetic measures?

The discussion on DPP4 is partly quite speculative and does not include studies that have investigated the association of DPP4 in the serum and BMD and/or fractures. I suggest focusing on apparent differences between DPP4 levels in the bone marrow as reported in this study vs. serum DPP4 levels, which in some studies did not correlate with BMD or fractures (e.g. Carbone et al. Osteoporosis International 2017).

How do the authors explain the difference between effects in the DMab vs. BP group on HbA1c and FPG? Both suppress coupling. Why does only DMab result in an improvement of diabetic parameters? This would rather suggest that DMab improves glucose homeostasis rather by direct actions of RANKL on e.g. hepatocytes, as shown by the group of Schett in 2013, or other yet unknown mechanisms, but not via its action on osteoclasts. If the coupling of osteoclast-derived DPP4 is critical for this action, one would assume that this effect should also have been present in the BP-treated diabetic group.

Minor points:

1. Figure legends should be described in more detail as especially concerning the statistical tests used. Furthermore, meaning of the error bar should be indicated (SD,SEM).
2. In Figure 4D, what does OS, ES and QS mean? Please mention in the Figure legend. Please add the negative probe as well. What is the meaning of the dotted line, please explain.
3. Figure 5 would benefit from explaining the different lines so it is clear at once, which line is which treatment group.
4. The abstract does not require LIF as a validated target, in my opinion. Focusing on DPP4 in the diabetes setting is enough to keep the story line straight.

Reviewer #3:

Remarks to the Author:

Megan M. et al, identified several coupling factors linking bone resorption to bone formation using RNA-seq and proteomics technique. The approach to identify factors involving in bone remodeling by examining genes downregulated by denosumab (DMAb) treatment, which results in both osteoclast numbers and bone formation, is basically interested, but identified factors has been already studied in animal model, therefore the novelty is very weak. Moreover, authors also tried to link osteoclast to type II diabetes by showing that blocking DPP4, which is secreted from osteoclast, by the treatment with DMAb decreased HbA1c and BMI. However, it is not clear what authors try to claim in this study. Do the authors want to try to find coupling factors during bone remodeling or find key mediators between bone remodeling and diseases related with energy metabolism such as type II diabetes? It may be better to study the detailed mechanism on how secreted factors from osteoclast regulate gene expression in osteoblast to support the study. Below are specific points

1. The description of background for this study in Introduction part is not sufficient. It needs to explain the relationship between osteoporosis and diabetes more in detail, because you mentioned the function of DMAb in glucose homeostasis in Figure 4 and 5.
2. In Figure 3, 5 genes including CREG2, DPP4, LIF, CST3 and CCBE1 were identified as osteoclast-specific genes suppressed by DMAb. Then, it should be suggested more biological meaning of this finding in DMAb-treated osteoporosis patients. Moreover, you showed the level of only DPP4 in serum. How about the level of the others?
3. I recommend that more detailed mechanism studies how DMAb treatment affect into not only inhibition of bone resorption but also bone formation by possibly utilizing and expanding more the

RNA-sequencing data.

4. The effect of DMAb in a diabetic cohort was shown in Figure 4 and 5. However, it is off the point of this study.
5. Throughout the manuscript, provide references which were reported previously.
6. In Figure 5, p-values in each factors (under graph) are not matched with values shown in graphs and Supplementary table 6.
7. In Figure 6, description about the function of RANKL into glucose homeostasis and the effect of glucose into osteoblast are not sufficient.

We would like to thank the Editors and Reviewers of *Nature Communications* for their thoughtful review of our manuscript entitled, "Identification of novel factors involved in the coupling of bone resorption and bone formation in humans reveals a new link between bone remodeling and energy metabolism" (NCOMMS-19-12791). We have addressed each of the points raised by the Reviewers and have either appropriately revised the manuscript or have provided a rationale for being unable to do so, as summarized below. All changes to the manuscript are highlighted in yellow.

Reviewer #1

The skeleton is a metabolically active endocrine organ. The best evidence for this comes from studies of FGF23, an osteocyte-derived phosphaturic hormone that also regulates renal 1-alpha hydroxylase activity, parathyroid hormone production, and cardiomyocyte hypertrophy. Preclinical murine models have suggested that the skeleton controls fuel metabolism via osteocalcin-regulated insulin and adiponectin production. However, any evidence for this or a similar bone-pancreatic endocrine axis in humans has been entirely unconvincing.

In the present study, Oursler, Khosla and colleagues provide the very first robust evidence of a skeletal - pancreatic endocrine axis that regulates fuel metabolism in humans. RANKL, an osteocyte-derived paracrine signal that promotes osteoclast-mediated bone loss, is inhibited by denosumab (D-mab) as an effective therapy for osteoporosis. During their studies deploying RNA-Seq, interrogating the transcriptome in bone from patients treated with D-Mab vs. placebo, the investigative team identified that skeletal expression of DPP4, a proteolytic inhibitor of the incretin GLP1, was down-regulated with D-Mab therapy. In situ hybridization identified that DPP4 was expressed in the osteoclast lineage. Consistent with this biology, bone marrow plasma and circulating levels of serum DPP4 were reduced in cohorts of patients treated with D-Mab ($p=0.02$) with concomitant increase in baseline circulating GLP1 ($p = 0.05$). Because there was no significant impact of D-Mab on glucose or insulin homeostasis in otherwise healthy (non-diabetic) women with osteoporosis, the investigative team studied a case-control cohort of patients with pre-diabetes or frank T2D treated with D-Mab vs. an aminobisphosphonate (aminoBP) vs. calcium+vitamin D for 1 year. Unlike D-Mab therapy, wherein osteoclast numbers are profoundly reduced (e.g. a pure "clastocide"), aminoBP therapy results in reduced osteoclast resorption activity in the setting of modest skeletal accrual of multinucleated osteoclast numbers. As compared to either aminoB or calcium plus vitamin D, HbA1c levels reduced in this setting, with concomitant reductions in fasting plasma glucose. BMI, and important contributor to insulin resistance in the setting of MetS and T2D, is also reduced by D-mab. Thus, the authors conclude that osteoclast-derived DPP4 as a RANKL-ligand regulated link between bone remodeling and fuel metabolism --thereby providing the very first robust evidence of a skeletal - pancreatic endocrine axis that regulates fuel metabolism in humans as noted above. I have only the following suggestions concerning this very important and enlightening manuscript:

- 1. Lipid profiles, particularly serum triglycerides, are important contributors to the cardiometabolic risk profile of MetS-T2D patients, often entrained to insulin sensitivity. Have the authors identified whether fasting triglycerides are reduced in D-mab treated cases vs. cohort controls (Ca+D, aminoBP)? Alternatively, were plasma HDL levels, often reciprocally related to serum Trigs, increased with D-Mab? Any such data would be a welcome addition, and provide additional evidence for their working model in Figure 6.*

Response: We agree that inclusion of lipid data would enhance our findings. We are unable to provide this data in the clinical study as lipids were inconsistently measured by the primary care physicians in that study, and our clinic does not routinely archive blood samples. However, we have now measured plasma total and HDL cholesterol and triglycerides, as well as calculated LDL cholesterol levels in the prospective cohort, with the limitation being that these were non-diabetic, healthy subjects. As shown in Supplementary Table 5 and now noted on page 11, changes in the above lipid parameters did not differ between the control and DMAb groups. Whether DMAb treatment would alter lipids, particularly triglyceride and HDL levels, in diabetic subjects remains to be investigated.

2. *Page 19, Methods: Because GLP1 varies so dramatically based upon prandial setting, etc., it would be very helpful to describe the conditions under which serum was drawn for studies of GLP1 and DPP4 in their human subjects. Were these drawn under fasting conditions?*

Response: All samples were drawn in the fasting state between 7 and 8 am (with the exception of the one subject who did not follow our instructions, and has now been excluded from this part of the analysis [see below]). This is now stated in the *Study protocol* on page 19.

3. *If the authors have calculated the HOMA-IR (Homeostatic model assessment of insulin resistance) and/or HOMA-beta for their relevant cohorts, this could be included in the online supplement (see also comment 5 below).*

Response: In responding to this point and calculating the HOMA-IR and HOMA- β values, we became aware of issues with 2 study participants that have now been addressed:

- a. Because the DMAb interventional study was originally designed to examine osteoclast-osteoblast coupling factors, we did not include fasting glucose as a screen. When we subsequently closely examined the glucose values for the HOMA-IR and HOMA- β calculations, we discovered that 1 of the 48 subjects did, in fact, have type 2 diabetes. Because clinical trials and studies in osteoporosis do not generally exclude patients with type 2 diabetes, we have continued to include this subject in the gene expression analyses, although we would note that excluding this subject did not change the results of that analysis. However, for the analyses specific to glucose metabolism (Fig. 4C, E, Supplementary Table 5), we have now excluded this subject due to her abnormal glucose values, and restricted the glucose homeostasis analysis to a non-diabetic cohort. This allows for a cleaner analysis of this cohort, and the possible effects of DMAb in patients with type 2 diabetes is specifically addressed in the subsequent clinical study.
- b. In calculating the HOMA-IR and HOMA- β parameters requested by Reviewer 1, we also discovered that a second subject had, contrary to our instructions, clearly eaten before her blood draw (high glucose, insulin, and GIP values that were elevated at baseline but normal at her 3 month visit). Because this would alter all of the parameters related to changes in glucose homeostasis with DMAb (but not her gene expression data, as that was obtained at the 3 month visit, when she did fast), we have also now excluded her from the glucose homeostasis analysis in this cohort (Fig. 4C, E, Supplementary Table 5).
- c. We have clearly delineated these issues in the *Study subjects* section on page 19. We should note that in the now entirely non-diabetic cohort and without the confounding post-meal value of the subject noted above, the increase in plasma GLP-1 following DMAb treatment is now clearly significant (Figure 4E, p-value changes from 0.051 to 0.035), entirely consistent with the significant reduction in serum DPP4 levels (Figure 4C).

With these caveats in mind, we now provide the HOMA-IR and HOMA- β data in our prospective cohort (Supplementary Table 5). Changes in these parameters did not differ significantly

between groups, although we should note that an important limitation of this analysis is that we do not have data on the effect of DMAb on postprandial glucose metabolism, including insulin secretion and action. Evidence to date suggests that DPP inhibitors have effects on both fasting and postprandial glucose metabolism and that the effects are, in fact, more marked in the postprandial period (refn 40). Thus, a prospective study that measures post-prandial β - and α -cell function together with glucose disposal is needed to further evaluate the mechanisms of the effects of DMAb on glucose homeostasis. We have added this point as a limitation of our study to the Discussion on pages 15-16.

4. *The authors may wish to briefly discuss the limitations of their case-cohort study design for the benefit of the Journal's broad audience.*

Response: We have added this to the Discussion on page 15-16, where we now acknowledge that our case-control study in diabetic patients has inherent limitations, specifically lack of blinding and potential confounders, and a rigorous randomized controlled trial examining the effects of DMAb on glycemic control in patients with type 2 diabetes mellitus and osteoporosis is clearly warranted. In addition, we have also noted the issue noted in point 3 (above) regarding lack of data on postprandial glucose metabolism.

5. *In their legend to Figure 6, the authors should mention that it remains to be shown that antagonism of RANKL by D-Mab regulates the insulin/glucagon ratio in humans. If, however, the authors have calculated the HOMA-IRs/HOMA-betas for their cohorts this could be included.*

Response: We have modified the legend accordingly, also noting that the insulin/glucagon ratio needs to be evaluated in detail not only under fasting, but also postprandial, conditions, given what is known about the mechanisms of action of DPP4 inhibitors (see point 3, above). We should also note that a very recent study by Bonnet and colleagues has provided further support for the model proposed in Figure 6, and we have now added a reference to this work in the Discussion on page 18, where we state: "Indeed, consistent with this hypothesis, Bonnet et al.⁵⁷ recently demonstrated that transgenic mice overexpressing RANKL have systemic insulin resistance and reduced glucose uptake in non-skeletal tissues (muscle, brain), but *increased* glucose uptake in bone (femur)." Thus, our findings and the model proposed in Figure 6 are entirely consistent with the new mouse data from Bonnet et al. with the further linkage in our work, in humans, of RANKL inhibition to DPP4 and GLP-1 (note the human data in the Bonnet paper is not related to glucose homeostasis, but rather to muscle function as assessed by grip strength in a non-randomized observational analysis). We have also modified Figure 6 to include the issue of RANKL inducing systemic insulin resistance; indeed, as noted in the response to Reviewer 2, point 6, and in the Discussion on pages 17-18, "these previously defined actions of RANKL on inducing insulin resistance may be linked to our finding that DMAb reduced circulating DPP4 and increased GLP-1 levels, in that both DPP4 inhibitors⁵⁴ and GLP-1⁵⁵ may also reduce insulin resistance."

6. *Unlike aminoBPs, the RANKL ligand antagonist Denosumab (D-mab) not only inhibits resorption but enables continued bone mineral accrual following therapeutic intervention beyond 2 years of therapy (plateaus on aminoBPs). This occurs even though markers of bone collagen synthesis remain reduced, a consolidation of matrix mineralization enabled by D-Mab but not aminoBPs. The molecular mechanisms contributing to this important biology that favorably "uncouples" bone mineral accrual from bone resorption is completely unknown. The authors data identified that skeletal expression of Wnt16, an important contributor to human (and mouse) bone mass as determined genetically, is increased with D-Mab (data supplement). Indeed, this discovery may be as important as the identification of*

the novel coupling factors emphasized. The authors may wish to very briefly comment on this observation in their discussion as well.

Response: This is indeed, a very interesting and important point that we had missed. Thank you for picking this up. We have now added this point to the Discussion on pages 16-17, where we now state: “Although we focused our analysis on osteoclast-derived secreted factors as candidate coupling factors, we should note an important osteoblast-derived gene, Wnt16, that was significantly upregulated (2.23-fold, P = 0.008; Supplementary Table 2) in the DMAb relative to placebo bone biopsies. Previous work by the Ohlsson group has shown that osteoblast-derived Wnt16 inhibits osteoclastogenesis and prevents cortical bone fragility in mice⁵¹, and SNPs in the Wnt16 gene have been associated with cortical bone thickness²⁵ and fracture risk⁵² in humans. The observed upregulation of Wnt16 expression by DMAb may provide an explanation for the finding from clinical trials that DMAb not only inhibits bone resorption, but enables continued bone mineral accrual following therapeutic intervention for up to 10 years of therapy.⁵³ This occurs even though markers of bone collagen synthesis remain suppressed, likely reflecting a consolidation of matrix mineralization enabled by DMAb, but not bisphosphonates.”

Minor comments:

7. *The observation that D-mab increases GLP1 while reducing DPP4 and BMI is intriguing. While GLP1 receptor agonists also cause weight loss, DPP4 antagonists are usually weight neutral in T2D. Do the authors have any thoughts as to the mechanisms of BMI reduction with D-mab therapy?*

Response: We have addressed this issue in the Discussion on page 16 as follows: “It will also be important to confirm our finding that DMAb treatment is associated with weight loss in a well powered, randomized, placebo-controlled prospective study. That said, it is well established that DPP4 inhibitors may raise GLP-1 concentrations but decrease net GLP-1 secretion.⁴¹⁻⁴³ This may limit the therapeutic efficacy of DPP4 inhibitors and prevent the degree of GLP-1 elevation necessary to modulate satiety and induce weight loss as observed with GLP-1 infusion or GLP-1 receptor agonist therapy.⁴⁴ Thus, it will be of interest to evaluate whether postprandial elevation of GLP-1 with DMAb exceeds that observed with DPP4 inhibitor treatment, thereby providing a potential explanation for the weight loss we noted following DMAb treatment.”

8. *Page 17, magnetic misspelled as “magentic.”*

Response: This has been corrected.

Reviewer #2

In the study by Weivoda et al. a single DMAb treatment was given as a biological probe to ablate osteoclasts in healthy postmenopausal women to identify potential secreted factors coupling bone resorption and bone formation. Based on RNA-sequencing of 15 bone samples per group they identified LIF, CREG2, CST3, CCBE1 and DPP4 as possible osteoclast-derived factors which are downregulated by DMAb, and might be involved in the coupling of bone resorption to bone formation. Circulating DPP4 was significantly reduced in DMAb treated participants when compared to the placebo-treated group. In addition, using a prospective study, DMAb treatment reduced HbA1c levels in type 2 diabetic patients as compared to patients either treated with bisphosphonates or calcium and vitamin D. Thus, this study highlights new coupling factors of bone resorption and formation in humans.

This study appears extremely well done in terms of obtaining high-quality RNA-seq data on bone and the collection of prospective cohort data. The data they have obtained in terms of bone remodeling genes is convincing and this kind of cohort is unique. Even though this is a beautiful proof-of-concept study, many of the results confirm what is already known and novel mechanistic insights are limited, even though novel candidates are proposed.

Response: We thank the Reviewer for the positive comments. In terms of novelty, we would note that although our study confirms LIF as a previously validated osteoclast-derived osteoblast coupling factor (in essence confirming the validity of our approach), we do describe, for the first time, several novel coupling factors *in humans*. In addition, as noted by Reviewer 1, our data are the first robust evidence in humans mechanistically linking bone remodeling to energy metabolism. We would suggest that providing deeper mechanistic insights in humans is extremely difficult, if not impossible, and further mechanistic insights need to utilize animal models.

1. *One exciting aspect is the more intricate analysis of DPP4. Bone marrow levels of DPP4 correlated with bone turnover gene sets. Was this also the case for serum DPP4 levels? What about GIP? As this is also a target of DPP4, was this parameter also assessed in the cohort and does it correlate with bone turnover? Furthermore, were DPP4 and GLP-1 levels also measured in the diabetic cohort? Did this correspond with HbA1c and/or bone turnover markers? In general, it would be interesting to show BMD (T-score) and bone turnover makers in the diabetic cohort and correlate these with the DPP4 targets, possibly even with HbA1c and FPG.*

Response:

- a. We have added the analysis of serum DPP4 levels as Supplementary Figure 5. Similar to bone marrow DPP4, serum DPP4 significantly correlated with osteoclast and osteoblast gene sets in the placebo participants, although the correlation coefficients were numerically smaller than those of bone marrow DPP4 with these gene sets. We now note this in the Results on page 10.
 - b. We now provide the GIP data (Supplementary Table 5). Although we did find a significant increase in plasma GLP-1 levels following DMAb treatment, changes in plasma GIP levels did not differ between groups. As noted in the response to Reviewer 1, point 3, this data needs to be interpreted with caution, as postprandial GIP levels were not assessed in our study, and these may have differed significantly between groups. Thus, further studies are needed to evaluate possible changes in postprandial GIP levels following DMAb treatment.
 - c. Since the diabetic cohort was a clinically treated cohort, we do not have DPP4 or GLP1 levels, or bone turnover markers in these subjects, as these were not obtained as part of their routine clinical care.
2. *Concerning the diabetic cohort, more patients in the DMab group were on life style intervention. Was that the reason for the more significant reduction of BMI as compared to the other groups? May this be a confounder for the analysis of diabetic measures?*

Response: We agree this is a potential limitation of our study, and as per Reviewer 1, point 4, we acknowledge these issues in the Discussion on pages 15-16, noting the need for a rigorous randomized controlled trial examining the effects of denosumab on glycemic control in patients with type 2 diabetes mellitus and osteoporosis to further validate our findings. That said, as noted in the Results on page 12, we do demonstrate that the change in HbA1c levels in DMAb-treated patients remained significant even after adjustment for changes in BMI (Supplemental Table 7).

3. *The discussion on DPP4 is partly quite speculative and does not include studies that have investigated the association of DPP4 in the serum and BMD and/or fractures. I suggest focusing on apparent differences between DPP4 levels in the bone marrow as reported in this study vs. serum DPP4 levels, which in some studies did not correlate with BMD or fractures (e.g. Carbone et al. Osteoporosis International 2017).*

Response: We have expanded the Discussion on pages 14-15 to include studies regarding DPP4 and BMD and BMD/fracture incidence. Specifically we address the papers published by Carbone et al. (refn 36) and Notsu et al. (refn 37). The study by Carbone et al. found no association of DPP4 activity with BMD or fracture incidence. The study by Notsu et al. found no association with BMD in Type 2 diabetic men; however, consistent with our findings, DPP4 levels correlated with bone turnover and they also found an association with vertebral fracture incidence. Based on our findings and this previous work, we acknowledge the need for further studies in understanding the role of DPP4 in overall bone metabolism.

4. *How do the authors explain the difference between effects in the DMab vs. BP group on HbA1c and FPG? Both suppress coupling. Why does only DMab result in an improvement of diabetic parameters? This would rather suggest that DMab improves glucose homeostasis rather by direct actions of RANKL on e.g. hepatocytes, as shown by the group of Schett in 2013, or other yet unknown mechanisms, but not via its action on osteoclasts. If the coupling of osteoclast-derived DPP4 is critical for this action, one would assume that this effect should also have been present in the BP-treated diabetic group.*

Response: The Reviewer raises an important point, and we now directly address this issue in the Discussion on page 16, where we state: “Our finding that patients treated with DMAb had significant reductions in HbA1c levels as compared to bisphosphonate-treated patients suggests important differences between effects of these anti-resorptive drugs on glucose homeostasis. These differences may arise from differing effects of these two drugs on osteoclasts. Thus, although both drugs inhibit bone resorption, long-term bisphosphonate therapy is associated with little or no reduction in the number of osteoclasts, but with abnormal appearing osteoclasts that include giant, hypernucleated, detached osteoclasts.⁴⁵ By contrast, DMAb markedly reduces osteoclast numbers on all bone surfaces.⁴⁶ Thus, bisphosphonates and DMAb may have differing effects on osteoclast DPP4 production, and this possibility warrants further investigation. In addition, as discussed below, DMAb may modulate effects of RANKL on glucose homeostasis involving the pancreatic islets⁴⁷⁻⁴⁹ or liver,⁵⁰ thereby exerting more beneficial effects on glucose metabolism as compared to bisphosphonates.” Finally, as we now note on pages 17-18, “these previously defined actions of RANKL on inducing insulin resistance may be linked to our finding that DMAb reduced circulating DPP4 and increased GLP-1 levels, in that both DPP4 inhibitors⁵² and GLP-1⁵³ may also reduce insulin resistance.” As noted in the response to Reviewer 1, point 5, we have also modified Figure 6 to better reflect these diverse effects of RANKL on glucose homeostasis based on previous mouse and now our human data.

Minor points:

1. *Figure legends should be described in more detail as especially concerning the statistical tests used. Furthermore, meaning of the error bar should be indicated (SD, SEM).*

Response: We apologize for the lack of detail and have gone over the figure legends to enhance their clarity and provide greater detail into the statistical methods employed for each figure.

2. *In Figure 4D, what does OS, ES and QS mean? Please mention in the Figure legend. Please add the negative probe as well. What is the meaning of the dotted line, please explain.*

Response: We have added definitions for OS (osteoid surface), ES (eroded surface), and QS (quiescent surface) to the figure legend. We have also defined the dotted lines, which are intended to show the separation of surfaces. We apologize for the lack of explanation, and thank the reviewer for making this point to improve the clarity of our figures. In addition, we have added a negative control image to demonstrate scientific rigor of our in situ hybridization (ISH) analysis.

3. *Figure 5 would benefit from explaining the different lines so it is clear at once, which line is which treatment group.*

Response: We have changed the colors of the lines and added a legend defining the groups to the image. We thank the reviewer for this suggested improvement to figure clarity.

4. *The abstract does not require LIF as a validated target, in my opinion. Focusing on DPP4 in the diabetes setting is enough to keep the story line straight.*

Response: The Reviewer does raise a valid point; however, our preference is to leave LIF in the abstract as a validated target as it provides a strong validation of our approach to identify coupling factors in humans.

Reviewer #3

Megan M. et al, identified several coupling factors linking bone resorption to bone formation using RNA-seq and proteomics technique. The approach to identify factors involving in bone remodeling by examining genes downregulated by denosumab (DMAb) treatment, which results in both osteoclast numbers and bone formation, is basically interested, but identified factors has been already studied in animal model, therefore the novelty is very weak. Moreover, authors also tried to link osteoclast to type II diabetes by showing that blocking DPP4, which is secreted from osteoclast, by the treatment with DMAb decreased HbA1c and BMI. However, it is not clear what authors try to claim in this study. Do the authors want to try to find coupling factors during bone remodeling or find key mediators between bone remodeling and diseases related with energy metabolism such as type II diabetes? It may be better to study the detailed mechanism on how secreted factors from osteoclast regulate gene expression in osteoblast to support the study. Below are specific points.

Response: We would respectfully disagree with the Reviewer's comments regarding lack of novelty, and would point out that Reviewers 1 and 2 agree with our position. Specifically, Reviewer 1 states that "the authors conclude that osteoclast-derived DPP4 as a RANKL-ligand regulated link between bone remodeling and fuel metabolism - thereby providing the very first robust evidence of a skeletal-pancreatic endocrine axis that regulates fuel metabolism in humans as noted above. I have only the following suggestions concerning this very important and enlightening manuscript"; and Reviewer 2 states that "this study highlights new coupling factors of bone resorption and formation in humans."

In terms of the second point raised above regarding the focus of our study, as we state in the Introduction (page 4), "we utilized a single DMAb treatment as a biological probe to pharmacologically ablate osteoclasts in post-menopausal women in order to identify potential osteoclast-derived factors that contribute to the coupling of bone resorption and bone formation in humans." This was the goal of the study, and is clearly laid out in the Introduction. As with any scientific endeavor, the data led us in an unexpected direction – specifically the link between bone and energy metabolism, and the paper endeavors to summarize our findings

both related to osteoclast-osteoblast coupling factors as well as our novel findings related to bone and energy metabolism.

1. *The description of background for this study in Introduction part is not sufficient. It needs to explain the relationship between osteoporosis and diabetes more in detail, because you mentioned the function of DMAb in glucose homeostasis in Figure 4 and 5.*

Response: As noted in the response to point 2, our goal was to identify osteoclast-derived factors that contribute to the coupling of bone resorption and bone formation in humans. As these studies unfolded, we were led into the studies relating bone to systemic energy metabolism. We would suggest that it would be disingenuous of us to suggest that we foresaw the unexpected directions that the study would take and state in the Introduction that we planned to study the relationship between bone and energy metabolism. Rather, in the Introduction, we state our original goals, and in the Discussion we discuss in great detail the links between bone and energy metabolism once our data led us in that specific direction.

2. *In Figure 3, 5 genes including CREG2, DPP4, LIF, CST3 and CCBE1 were identified as osteoclast-specific genes suppressed by DMAb. Then, it should be suggested more biological meaning of this finding in DMAb-treated osteoporosis patients. Moreover, you showed the level of only DPP4 in serum. How about the level of the others?*

Response: We would point out that the biology of LIF as an osteoclast-derived osteoblast coupling factor has been examined in detail (see Discussion on page 13), CST3 has been shown to induce osteoblast differentiation *in vitro*, and CREG2 and CCBE1 have yet to be assessed for potential roles in bone metabolism. In performing the Olink proteomics assay, we were setting out to identify potential factors that were differentially expressed at the protein level in placebo vs. DMAb patients. Our intent was not to validate our mRNA analysis. Of the factors identified by mRNA analysis, both DPP4 and CST3 were reduced at the protein level. LIF was assessed in the Olink panel; however, the levels were below the level of detection for the majority of patients. CREG2 and CCBE1 are not available in the Olink panels. In the paper, however, we chose to focus on the biology of DPP4, as this factor came up repeatedly in our mRNA analysis, along with protein analysis, and correlation with bone gene sets. In addition, this finding provided a novel, unexpected link between bone and energy metabolism.

3. *I recommend that more detailed mechanism studies how DMAb treatment affect into not only inhibition of bone resorption but also bone formation by possibly utilizing and expanding more the RNA-sequencing data.*

Response: Reviewer 1 also commented on this issue, so please see our response to Reviewer 1, point 6. We have now added this point to the Discussion on pages 16-17, where we now state: "Although we focused our analysis on osteoclast-derived secreted factors as candidate coupling factors, we should note an important osteoblast-derived gene, Wnt16, that was significantly upregulated (2.23-fold, P = 0.008; Supplementary Table 2) in the DMAb relative to placebo bone biopsies. Previous work by the Ohlsson group has shown that osteoblast-derived Wnt16 inhibits osteoclastogenesis and prevents cortical bone fragility in mice⁵¹, and SNPs in the Wnt16 gene have been associated with cortical bone thickness²⁵ and fracture risk⁵² in humans. The observed upregulation of Wnt16 expression by DMAb may provide an explanation for the finding from clinical trials that DMAb not only inhibits bone resorption, but enables continued bone mineral accrual following therapeutic intervention for up to 10 years of therapy.⁵³ This occurs even though markers of bone collagen synthesis remain suppressed, likely reflecting a consolidation of matrix mineralization enabled by DMAb, but not bisphosphonates."

4. *The effect of DMAb in a diabetic cohort was shown in Figure 4 and 5. However, it is off the point of this study.*

Response: We would refer the Reviewer to our responses to points 2 and 3. Rather than being “off point”, we were simply following the directions mandated by the results of our analyses.

5. *Throughout the manuscript, provide references which were reported previously.*

Response: Our paper contains 67 references. To our knowledge, we have acknowledged virtually all of the previous work relevant to our findings, but would welcome any suggestions from the Reviewer regarding key references we may have missed.

6. *In Figure 5, p-values in each factors (under graph) are not matched with values shown in graphs and Supplementary table 6.*

Response: We apologize for the misunderstanding. The p-values listed under the graphs in Figure 5 are specific to our first analysis of the patient cohort, in which we did not adjust for change in BMI. In Supplementary Table 6 (now Supplementary Table 7), we performed the analysis with adjustment for change in BMI, to confirm that the improvement in HbA1c levels was not an artifact of improved BMI in DMAB treated patients. We have added an additional explanation to the table legend to make this clearer.

7. *In Figure 6, description about the function of RANKL into glucose homeostasis and the effect of glucose into osteoblast are not sufficient.*

Response: Figure 6 is intended to simply be a working model to help stimulate future studies, where we do summarize the key literature and have now also added a recent reference that very much supports the model proposed in the Figure. Specifically, as noted in the response to Reviewer 1, point 5, we now note that a very recent study by Bonnet and colleagues has provided further support for the model proposed in Figure 6, and we have added a reference to this work in the Discussion on page 17, where we now state: “Indeed, consistent with this hypothesis, Bonnet et al.⁴⁸ recently demonstrated that transgenic mice overexpressing RANKL have systemic insulin resistance and reduced glucose uptake in non-skeletal tissues (muscle, brain), but increased glucose uptake in bone (femur).” In addition As noted in the response to Reviewers 1 and 2, we have also modified Figure 6 to reflect better reflect these diverse effects of RANKL on glucose homeostasis based on previous mouse and now our human data.

We are submitting our extensively revised manuscript for further consideration and hope it is now acceptable for publication in *Nature Communications*.

Sincerely,

Reviewers' Comments:

Reviewer #1:

Remarks to the Author:

Excellent revision, no additional comments.

Reviewer #2:

Remarks to the Author:

The authors have responded comprehensively to my comments/suggestions and have included more data and discussion points. I believe the discussion is now more balanced and well-rounded. I have no major comments left expect for one minor suggestions: in the discussion paragraph on possible differences between Dmab and BP, the authors may want to consider adding a sentence about the mineralization accrual as mentioned by reviewer 1 that is encouraged by Dmab but not by BP. That practically also leads over to their next discussion point namely osteoblasts and Wnt16. Thus, the differences between Dmab and BP could be threefold: 1) differences on OC, 2) differences on OB, 3) differences in organismic glucose handling.

Reviewer #3:

Remarks to the Author:

Most of concerns are well addressed.

One thing that we are not yet clear is about the coupling role of CREG2, DPP4, and CCBE1 because there is lack of molecular mechanism in this study.

Therefore, we suggest to revise the arrow of "DPP4 to osteoblasts" and delete "CREG2 and CCBE1" from coupling factor in figure 6.

We would like to thank the Editors and Reviewers of *Nature Communications* for their thoughtful consideration of our manuscript entitled, "Identification of novel factors involved in the coupling of bone resorption and bone formation in humans reveals a new link between bone remodeling and energy metabolism" (NCOMMS-19-12791). We are, of course, delighted by the provisional acceptance of our manuscript and are revising it further based on the additional comments. In this letter we provide our response to the remaining reviewer comments.

Reviewer 1

Excellent revision, no additional comments.

Response: Thank you.

Reviewer 2

The authors have responded comprehensively to my comments/suggestions and have included more data and discussion points. I believe the discussion is now more balanced and well-rounded. I have no major comments left expect for one minor suggestion: in the discussion paragraph on possible differences between Dmab and BP, the authors may want to consider adding a sentence about the mineralization accrual as mentioned by reviewer 1 that is encouraged by Dmab but not by BP. That practically also leads over to their next discussion point namely osteoblasts and Wnt16. Thus, the differences between Dmab and BP could be threefold: 1) differences on OC, 2) differences on OB, 3) differences in organismic glucose handling.

Response: A good suggestion, and we have added a sentence on this point in the Discussion on page 16.

Reviewer 3

Most of concerns are well addressed. One thing that we are not yet clear is about the coupling role of CREG2, DPP4, and CCBE1 because there is lack of molecular mechanism in this study. Therefore, we suggest to revise the arrow of "DPP4 to osteoblasts" and delete "CREG2 and CCBE1" from coupling factor in figure 6.

Response: The Reviewer raises a valid point and in response, we have modified Figure 6 as noted in the legend to that Figure: "RANKL signaling in osteoclasts induces (directly or indirectly) the expression of key coupling factors, identified in our human study as *LIF*, *CREG2*, *CST3*, *CCBE1*, and *DPP4* (note that *LIF* and *CST3* are known to stimulate bone formation and thus are likely osteoclast-osteoblast coupling factors [solid arrows],^{19,20,27} whereas *DPP4*, *CREG2*, and *CCBE1* are potential osteoclast-derived coupling factors identified by our study that require further characterization [dashed arrows])." We believe this addresses the Reviewer's concern without deleting the potential coupling factors we identified from the Figure; rather we now clearly separate the established coupling factors from potential coupling factors that need further mechanistic studies, as the Reviewer notes.

We are submitting our further revised manuscript for further consideration and hope it is now acceptable for publication in *Nature Communications*.

Sincerely,